# A panel of induced pluripotent stem cells from chimpanzees: a resource for comparative functional genomics

Irene Gallego Romero[1*†], Bryan J Pavlovic[1†], Irene Hernando-Herraez[2], Xiang Zhou[3], Michelle C Ward[1], Nicholas E Banovich[1], Courtney L Kagan[1], Jonathan E Burnett[1], Constance H Huang[1], Amy Mitrano[1], Claudia I Chavarria[1], Inbar Friedrich Ben-Nun[4‡], Yingchun Li[5,6], Karen Sabatini[4,6,7], Trevor R Leonardo[4,6,7], Mana Parast[5,6], Tomas Marques-Bonet[2,8], Louise C Laurent[6,7], Jeanne F Loring[4], Yoav Gilad[1*]

[1]Department of Human Genetics, University of Chicago, Chicago, United States; [2]Institut de Biologia Evolutiva (CSIC/UPF), Parc Recerca Biomèdica de Barcelona, Barcelona, Spain; [3]Department of Biostatistics, University of Michigan, Ann Arbor, United States; [4]Center for Regenerative Medicine, Department of Chemical Physiology, The Scripps Research Institute, La Jolla, United States; [5]Department of Pathology, University of California San Diego, San Diego, United States; [6]Sanford Consortium for Regenerative Medicine, La Jolla, United States; [7]Department of Reproductive Medicine, University of California San Diego, San Diego, United States; [8]Centro Nacional de Análisis Genómico (CNAG-CRG), Barcelona, Spain

*For correspondence: ireneg@ uchicago.edu (IGR); gilad@ uchicago.edu (YG)

†These authors contributed equally to this work

Present address: ‡Lonza Walkersville, Inc., Walkersville, United States

Competing interests: The authors declare that no competing interests exist.

**Abstract** Comparative genomics studies in primates are restricted due to our limited access to samples. In order to gain better insight into the genetic processes that underlie variation in complex phenotypes in primates, we must have access to faithful model systems for a wide range of cell types. To facilitate this, we generated a panel of 7 fully characterized chimpanzee induced pluripotent stem cell (iPSC) lines derived from healthy donors. To demonstrate the utility of comparative iPSC panels, we collected RNA-sequencing and DNA methylation data from the chimpanzee iPSCs and the corresponding fibroblast lines, as well as from 7 human iPSCs and their source lines, which encompass multiple populations and cell types. We observe much less within-species variation in iPSCs than in somatic cells, indicating the reprogramming process erases many inter-individual differences. The low within-species regulatory variation in iPSCs allowed us to identify many novel inter-species regulatory differences of small magnitude.

## Introduction

Comparative functional genomic studies of humans and other primates have been consistently hindered by a lack of samples (*Gallego Romero et al., 2012*). In spite of their clear potential to inform our understanding of both human evolution and disease, practical and ethical concerns surrounding working with non-human primates have constrained the field to using a limited set of cell types collected in a non-invasive or minimally invasive manner, primarily lymphoblastoid cell lines (LCLs) and fibroblasts. Comparative studies of any other primate tissue have been limited to using post-mortem (typically frozen) materials, thereby precluding most experimental manipulation and yielding primarily observational insights (see, e.g., *Blekhman et al., 2008*; *Blekhman et al. 2010*; *Brawand et al., 2011*).

An alternative has been to use model organisms in an attempt to recapitulate inter-primate regulatory differences. The typical approach involves the introduction of sequences of evolutionary

**eLife digest** Comparing the genomes of different species can reveal how they are related to one another. Such comparative studies can also reveal how genomes are modified in species-specific ways to regulate gene activity. The genomes of humans and chimpanzees are very similar in sequence. It is therefore likely that differing patterns of gene regulation underlie many of the differences observed between the two species. However, only a few kinds of chimpanzee cell that can be grown in the laboratory are available for research; this lack of samples has limited the ability of researchers to perform such comparative studies.

One way around this problem is to use induced pluripotent stem cells (or iPSCs). IPSCs are created by exposing mature cells—for example, skin cells—to conditions and molecules that convert them into an embryonic-like state. This state—called 'induced pluripotency'—allows the cells to be coaxed into becoming many different cell types that can be grown in the laboratory. But it is more difficult to establish high quality iPSCs from chimpanzees than it is from humans or mice.

Gallego Romero, Pavlovic et al. have now addressed this problem by creating iPSCs from skin cells taken from seven healthy chimpanzees. These cell lines were then analysed and compared to each other and to seven iPSC lines created from human cells. The chimpanzee iPSC lines were found to be much more similar to each other than the mature cells that were used to make them. Similar results were also observed for the human iSPCs, which likely reflects the conserved changes that take place when the genomes of mature cells are reprogrammed to pluripotency.

This high level of similarity between iPSCs from different individuals of the same species allowed Gallego Romero, Pavlovic et al. to discover many subtle differences in gene regulation between chimpanzees and humans. For example, over 4500 genes were found to be expressed differently in human and chimpanzee iPSCs, and over 3500 genomic regions had different patterns of certain DNA modifications that can help to regulate gene expression.

These newly created chimpanzee iPSC lines represent a valuable resource for comparative studies of gene regulation. In the future, this resource could help researchers to identify further differences in gene regulation between closely related primate species.

interest into a model system, and then searching for spatial or temporal differences in gene expression that can be ascribed to the introduced sequence (*Enard et al., 2009*; *Cotney et al., 2013*). This is a difficult and challenging approach and, perhaps as a result, there are still only a handful of well-described examples of human-specific regulatory adaptations in primates (*Prabhakar et al., 2008*; *McLean et al., 2011*) and even fewer cases where the underlying regulatory mechanisms have been resolved (*Rockman et al., 2005*; *Pollard et al., 2006*). While these studies are useful and often informative, they also entail assumptions of functional conservation between the model system and the species of interest that may not necessarily be true (*Gallego Romero et al., 2012*).

Induced pluripotent stem cells (iPSCs) can provide a viable means of circumventing these concerns and limitations, at least with respect to the subset of phenotypes that can be studied in in vitro systems. Reprogramming somatic cell lines to a stable and self-sustaining pluripotent state (*Takahashi and Yamanaka, 2006*; *Takahashi et al., 2007*) has become routine practice for human and murine cell lines, but extension to other animals, especially non-human primates, is not yet widespread despite some exceptions (e.g., *Ezashi et al., 2009*; *Ben-Nun et al., 2011*; *Nagy et al., 2011*; *Marchetto et al., 2013b*). Instead, the broadest application of iPSCs to date has been the generation of lines derived from patients suffering from a variety of genetic disorders (*Cohen and Melton, 2011*; *Israel et al., 2012*; *Liu et al., 2012*; *Merkle and Eggan, 2013*; *Wang et al., 2014*), with the dual aims of providing a deeper understanding of disease phenotypes and developing new therapeutic avenues. These cell lines have been shown to display in vitro properties corresponding to relevant patient phenotypes observed in vivo, both as iPSCs and when differentiated into other pertinent cell types, supporting their utility in clinical applications; more generally, these properties also highlight the tantalizing flexibility of iPSCs as a means of exploring developmental and cell lineage determination pathways.

Thus, the development of an iPSC-based system for comparative genomic studies in primates will allow us to compare regulatory pathways and complex phenotypes in humans and our close evolutionary relatives using appropriate models for different tissues and cell types. This will be

a powerful resource with which to examine the contribution of changes in gene regulation to human evolution and diversity. To demonstrate the validity of this approach, we have generated a panel of 7 chimpanzee iPSC lines that are fully characterized and comparable to human iPSC lines in their growth and differentiation capabilities.

## Results

We generated a panel of iPSC lines from seven chimpanzees through electroporation of episomal plasmids expressing *OCT3/4* (also known as *POU5F1*), *SOX2*, *KLF4*, *L-MYC*, *LIN28*, and an shRNA targeting *TP53* (*Okita et al., 2011*), as well as an in vitro-transcribed *EBNA1* mRNA transcript (*Howden et al., 2006*; *Chen et al., 2011*) that promotes increased exogenous vector retention in the days following electroporation. Our chimpanzee panel is comprised of seven healthy individuals (4 female, 3 male, further details on these individuals are given in *Supplementary file 1*) ranging from 9 to 17 years old. Fibroblasts from 5 of the 7 individuals were purchased from the Coriell Institute for Medical Research, while the remaining two (C6, C7) were derived from 3 mm skin punch biopsies directly collected from animals at the Yerkes Primate Research Center of Emory University (see 'Materials and methods'). All chimpanzee iPSC lines described in this publication are available fully and without restrictions to other investigators upon request to the corresponding authors.

### Characterizing the chimpanzee iPSCs

The chimpanzee iPSC lines closely resemble human iPSC lines in morphology (*Figure 1A*; all images shown in main text are from chimpanzee line C4. Similar images of the other lines are available as *Figure 1—figure supplements 1–5*). All lines could be maintained in culture for at least 60 passages without loss of pluripotency or self-renewal capability using standard iPSC culture conditions, both on mouse embryonic fibroblast (MEF) feeder cells and in feeder-free conditions. The genomes of all our lines appeared to be cytogenetically stable; all exhibited normal karyotypes after reprogramming and more than 15 passages in culture, ruling out the presence of gross chromosomal abnormalities (*Figure 1B*, *Figure 1—figure supplement 1*).

We confirmed nuclear expression of *OCT3/4*, *SOX2* and *NANOG* in all lines by immunocytochemistry (*Figure 1C*; *Figure 1—figure supplement 2*). The pluripotent cells also express the surface antigens Tra-1-81 and SSEA4, while cells collected from the center of differentiating colonies expressed SSEA1 at levels comparable to differentiating colonies of human iPSC lines (*Figure 1—figure supplement 3*). To confirm that the observed expression of pluripotency-associated genes is of endogenous origin, we performed qPCR with primers designed to specifically amplify the endogenous *OCT3/4*, *SOX2*, *NANOG* and *L-MYC* transcripts (*Figure 1D*; all PCR primers used in this work are listed in *Supplementary file 2*). Indeed, we found no evidence of exogenous gene expression after 10 passages (*Figure 1—figure supplement 4*), and no traces of genomic integration or residual episomal plasmid retention after 15 passages (*Figure 1E*). These observations indicate that self-renewal in our chimpanzee iPSC lines is maintained solely through endogenous gene expression.

To confirm pluripotency and test the differentiation capabilities of our lines, we performed a number of assays. First, we generated embryoid bodies from all 7 chimpanzee iPSC lines and assayed their ability to spontaneously differentiate into the three germ layers by immunocytochemistry. All lines spontaneously gave rise to tissues from the three germ layers (*Figure 2A*; *Figure 2—figure supplement 1*). Second, we carried out directed differentiations to hepatocytes and cardiomyocytes in a subset of the lines using previously published protocols (see 'Materials and methods', *Figure 2—figure supplement 2* and *Video 1*). Third, we performed teratoma formation assays in four of the lines using Fox Chase SCID-beige and CB17.Cg-*Prkdc^scid^Lyst^bg-J*/Crl immunodeficient male mice. All four iPSC lines were capable of generating tumours in mice, and all tumours examined contained tissues of endodermal, ectodermal and mesodermal origins (*Figure 2B*, *Figure 2—figure supplement 3*). To confirm the chimpanzee origin of these tissues, we extracted and performed Sanger sequencing on mitochondrial DNA from the tumours (*Figure 2—figure supplement 4*).

Finaly, we characterized pluripotency in our lines through PluriTest, a bioinformatic classifier that compares the gene expression profiles of new lines to those obtained from a reference set of over 400 well-characterized human pluripotent and terminally differentiated lines (*Müller et al., 2011*), modified to accommodate data from both species. All chimpanzee lines have PluriTest pluripotency scores greater than the pluripotency threshold value of 20 (*Figure 3A*, *Supplementary file 1*). We also

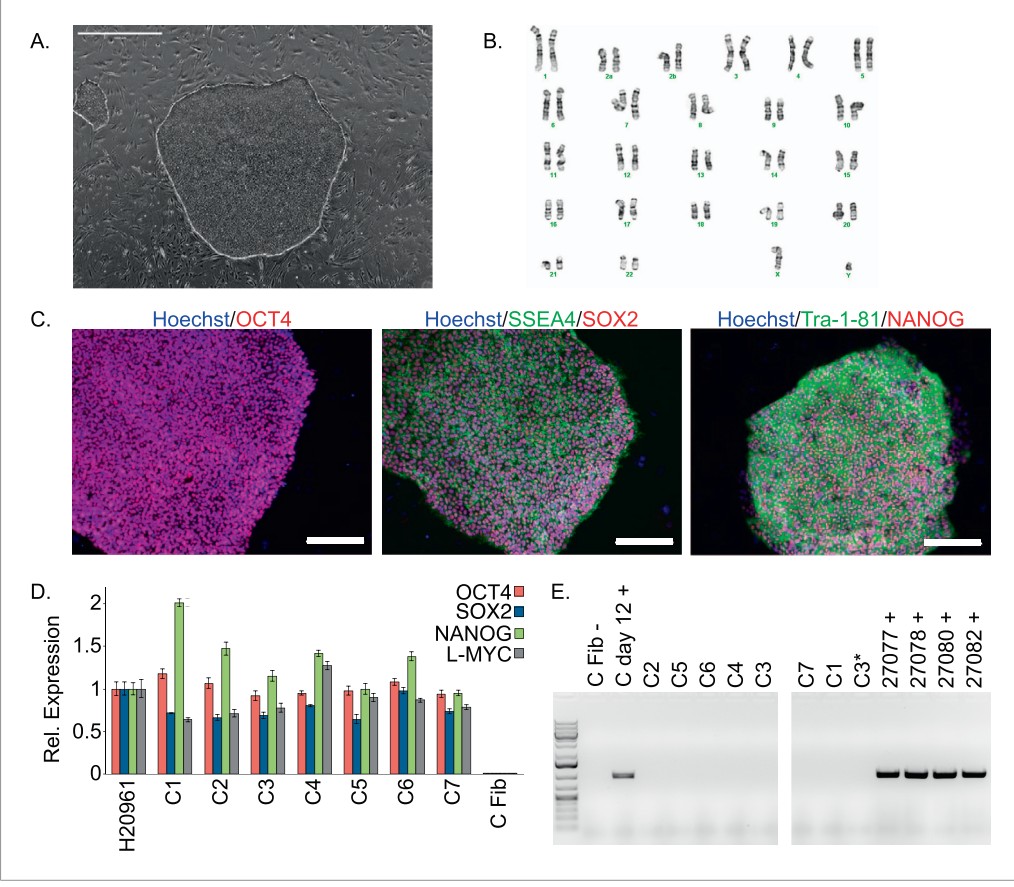

**Figure 1**. Characterization of chimpanzee induced pluripotent stem cell (iPSC) lines. (**A**) Phase contrast image of representative chimpanzee iPSC line. Scale bar: 1000 μm. (**B**) Representative karyotype from chimpanzee iPSC line after >15 passages, showing no abnormalities. (**C**) ICC staining of iPSC lines with antibodies for pluripotency markers as indicated. Scale bar: 200 μm. (**D**) Quantitative PCR testing for expression of endogenous pluripotency factors in all 7 chimpanzee iPSC lines. Line H20961 is a male human iPSC line generated in-house used as reference. (**E**) PCR gel showing an absence of exogenous episomal reprogramming factors in all 7 chimpanzee iPSC lines. All PCRs were carried out on templates extracted from passage >15 with the exception of C3651*, which is from passage 2. Fib—is a negative fibroblast control (from individual C8861) prior to transfection, day 12 + is a positive control 12 days after transfection, 27,077 + to 27,082 + are the plasmids used for reprogramming.

The following figure supplements are available for figure 1:

**Figure supplement 1**. Karyotypes for the 6 chimpanzee iPSC lines not shown in main text figures, generated after >15 passages in culture.

**Figure supplement 2**. ICC staining of the 6 chimpanzee iPSC lines not shown in main text figures with antibodies for pluripotency markers as indicated.

**Figure supplement 3**. ICC staining showing SSEA1 expression in chimpanzee iPSC culture plates, clearly distinct from NANOG expression.

**Figure supplement 4**. Melt curves showing a lack of exogenous reprogramming gene expression in episomally reprogrammed chimpanzee iPSCs after >10 passages.

**Figure supplement 5**. Exogenous gene expression in retrovirally reprogrammed chimpanzee iPSCs after various passages.

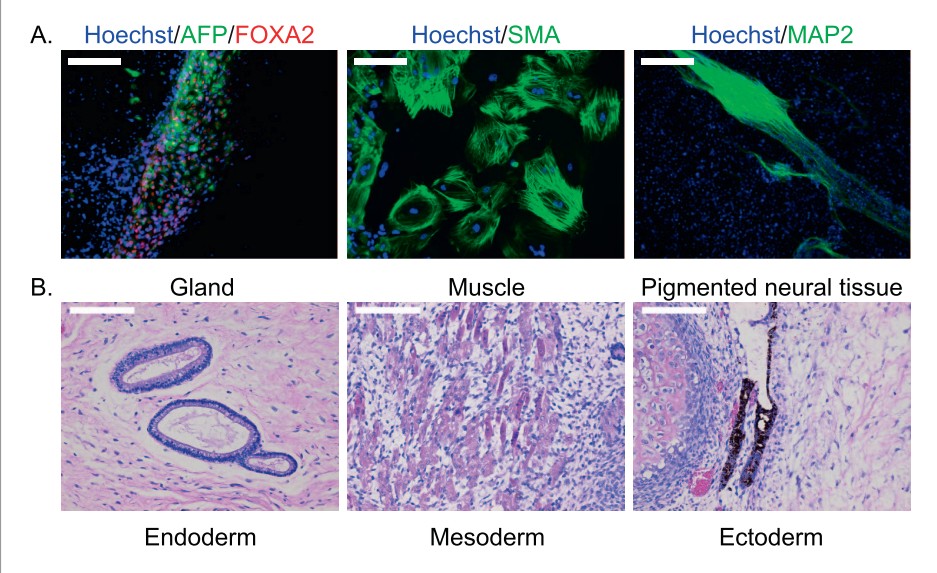

**Figure 2**. (**A**) ICC staining of differentiated embryoid bodies with antibodies for the three germ layers as indicated. Scale bar: 200 μm. (**B**) Histological staining of teratomas derived from iPSC line C4955, showing generation of tissues from all three germ layers. Scale bar: 100 μm.

The following figure supplements are available for figure 2:

**Figure supplement 1**. ICC staining of differentiated embryoid bodies derived from the 6 chimpanzee iPSC lines not shown in main text figures, with antibodies for the three germ layers as indicated.

**Figure supplement 2**. ICC staining of directly differentiated hepatocytes from line C2, with antibodies as indicated.

**Figure supplement 3**. Histological staining of teratomas derived from three additional chimpanzee iPSC lines, showing generation of tissues from all three germ layers.

**Figure supplement 4**. Sequencing traces from teratomas generated from chimpanzee iPSC lines for the mitochondrial genes *12S* (C3649, C4955) and *cytb* (C8861, C40210).

calculated PluriTest novelty scores for all samples. In human PSCs, novelty values above 1.67 are suggestive of chromosomal duplications or expression of differentiation-associated genes. Human PSCs with high novelty scores are typically either difficult to maintain and expand in culture (because they differentiate spontaneously at a high rate), or cannot be consistently differentiated to all three germ layers. All of our chimpanzee lines had novelty scores above the 1.67 threshold (*Figure 3B*). However, in contrast to human PSCs with high novelty scores, our chimpanzee lines can be both easily maintained in culture and differentiated into all three germ layer lineages, as demonstrated by the embryoid body and teratoma assays detailed above. We thus hypothesize that the observed high novelty scores are likely driven by inter-species gene regulatory differences that the PluriTest assay, which was trained exclusively on human samples, interpreted as abnormal gene expression.

## Interspecies analysis of gene expression and methylation data from iPSCs

To better examine gene expression and regulatory differences between human and chimpanzee iPSCs, we generated genome-wide RNA-sequencing and DNA methylation data (see 'Materials and methods') from all chimpanzee iPSC lines, as well as from 7 human iPSC lines also generated and validated in our laboratory. While all of the chimpanzee iPSCs were derived from fibroblast cell lines (*Supplementary file 1*), the human iPSCs were derived from both fibroblasts and immortalised LCLs from Caucasian and Yoruba individuals (see *Supplementary file 1* for additional details). We designed

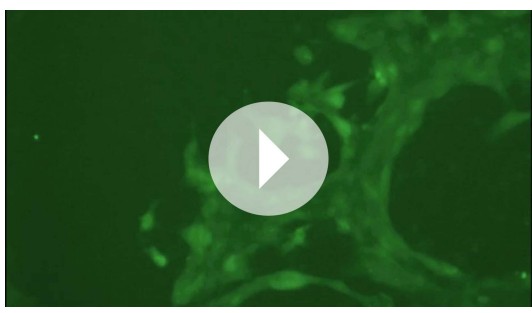

**Video 1.** Calcium transient flux in and out (GFP labelled) and contractility of directly differentiated cardiomyocytes from chimpanzee iPSC line C7.

the comparative study this way in order to demonstrate that regulatory differences between human and chimpanzee iPSCs cannot be explained by technical differences due to culturing conditions or the cell type of the somatic precursor cells used for reprograming.

To prevent biases due to genetic divergence between the two species, we chose to restrict our gene expression analyses to a curated set of genes with one-to-one orthology between humans and chimpanzees (*Blekhman et al., 2010*; *Blekhman, 2012*). Following assessment of quality control metrics (see 'Materials and methods'), we obtained normalised RPKM estimates for 12,171 genes that were expressed in at least 4 iPSC lines from either one of the species (see 'Materials and methods'). We similarly restricted our DNA methylation analyses to a set of 335,307 high quality probes with a high degree of sequence conservation between humans and chimpanzees (as in *Hernando-Herraez et al., 2013*; see 'Materials and methods').

To examine broad patterns in the data, we used principal component analysis (PCA). We observed clear and robust separation of human and chimpanzee iPSC lines along the first principal component (PC) in both the gene expression and DNA methylation data (*Figure 4A,B*; regression of PC1 by species; $p < 10^{-13}$ for the expression data; $p < 10^{-12}$ for the DNA methylation data). Within the human samples, PC2 appears to be driven by ethnicity, as we observe all Caucasian samples consistently clustering together despite their different cell types of origin ($p = 0.005$ for the association between PC2 and human ethnicity in the expression data, $p = 0.044$ in the DNA methylation data).

We then analysed regulatory differences between the species by first focusing on the gene expression data. At an FDR of 1%, we identified 4609 genes (37.9%) as differentially expressed (DE) between the iPSCs of the two species (*Supplementary file 3*; see 'Materials and methods' for details). The majority of DE genes do not exhibit large inter-species fold-change differences in expression levels (*Figure 4—figure supplements 1, 2*). An analysis of functional annotation of the DE genes reveals that no Gene Ontology Biological Process terms (GO BP; *Ashburner et al., 2000*) are significantly overrepresented among these genes at an FDR of 5% (*Supplementary file 4*), although we identified 123 overrepresented terms if we limit our analysis to the 546 genes with absolute $\log_2$ fold-change difference >2 (*Supplementary file 4*). Additionally, we tested for concordance between our list of DE genes and a list of 2730 genes that were previously classified as DE between human and non-human primate iPSC lines (*Marchetto et al., 2013b*). Given our stringent approach to consider orthologous genes, only 2081 (76%) genes could be analysed across the two studies. Of these, 1495 genes are detectably expressed in our lines, and 1079 (72.2%) are classified as DE between the species in both data sets (a highly significant enrichment; $\chi^2$ $p < 10^{-16}$). Expression trends within these DE genes are in the same direction in both data sets in 1060 of cases (98.24%).

Next, we used a similar approach to identify differentially methylated (DM) probes and regions between the iPSCs of both species (see 'Materials and methods'). We identified 63,791 probes that are DM between the two species at an FDR of 1%, 26,554 of which have a mean intergroup $\beta$ difference $\geq 0.1$, our arbitrary effect size threshold for retaining probes for DM region (DMR) identification and downstream analyses. Of these, 10,460 probes could be further grouped into 3529 regions of 2 or more DM probes within 1 kb, which we designated DMRs; (*Supplementary file 5*); the numbers of probes and regions identified as DM at a range of mean interspecies $\beta$ thresholds are given in *Supplementary file 6*.

In order to consider the DNA methylation and gene expression data jointly, we focused on a subset of 2348 DMRs that could be associated with a single Ensembl gene. Overall, these DMRs were associated with 2141 genes, of which 1350 were also detectably expressed in the iPSCs, and 558 (41.3%) were classified as DE between the species, a slightly higher proportion than expected by chance alone ($p = 0.1$). We further classified the DMRs as either 'promoter', 'genic' or 'mixed' depending on their position relative to annotated gene transcripts (see 'Materials and methods'). The overall set of DMRs, as well as genic DMRs, are significantly associated with 4 and 79 GO BP

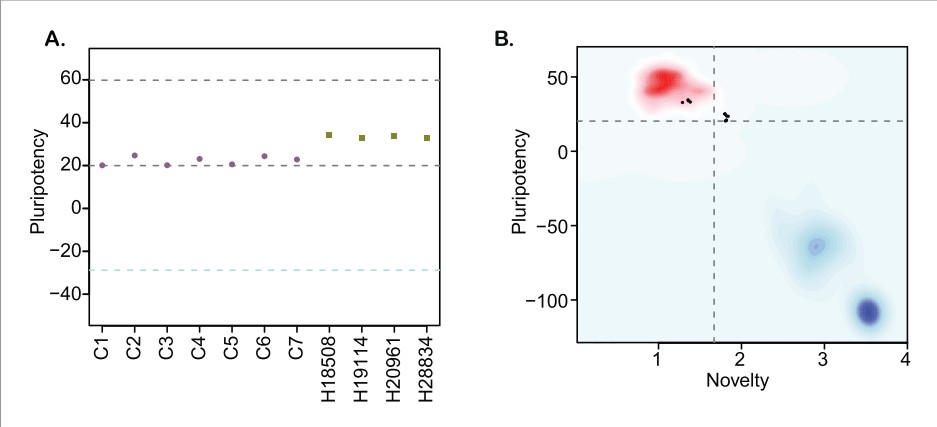

**Figure 3**. (**A**) PluriTest pluripotency scores in the 7 chimpanzee lines and 4 human reference iPSC lines. Purple circles denote chimpanzees; yellow squares, humans. (**B**) PluriTest results after removal of probes not mapping to the chimpanzee genome. All samples in the top left quadrant are human and have satisfactory pluripotency and novelty scores. Samples in the top right quadrant correspond to our chimpanzee iPSC panel, and have consistently high pluripotency yet high novelty scores.
The following figure supplement is available for figure 3:

**Figure supplement 1**. The effects of probe sub-setting in PluriTest pluripotency score calculations.

terms respectively (FDR < 5%), including terms related to neurogenesis and skeletal system development. Enrichment of several terms related to neurogenesis and skeletal system development is likewise marginally significant amongst promoter and mixed DMRs (*Supplementary file 7*). However, the subset of inter-species DE genes that are also associated with DMRs are not significantly enriched with annotation for any GO BP or MF terms.

## Comparative histone modification data

We used ChIP-seq to characterize the genome-wide distribution of two types of histone modifications (H3K27me3 and H3K27ac) in three of our chimpanzee iPSCs (see 'Materials and methods'). We compared the chimpanzee data to histone modification data from three human iPSC lines from the Roadmap Epigenomics project (*Figure 5*). To do so, we downloaded raw sequence files from GEO and processed data from both species using the same pipeline (see 'Materials and methods'). We identified ChIP-seq peaks using MACS or RSEG, as appropriate, and accounted for differences in genome sequence between the species as well as for incomplete power to identify peaks across species (see 'Materials and methods'). To relate the ChIP-seq data to genes (and integrate over data from all peaks that are in proximity to a given gene), we then generated enrichment ChIP scores for a set of previously defined 26,115 orthologous transcription start sites (TSSs, from *Zhou et al., 2014*). The enrichment score (see 'Materials and methods' for details, also *Supplementary file 8*), reflects the ratio of mapped ChIP-seq read counts across all peaks within a 4 kb window centred on an orthologous TSS, relative to the genome-wide read count average after adjusting expectations based on the input control sample. We chose to classify as 'enriched' any region where the mean enrichment score across all three individuals in the species was larger than 1. This cut-off is arbitrary, but we confirmed that our qualitative results are robust by additionally testing enrichment cut-offs of 2, 5, and 10.

Using this approach, we first examined genome-wide patterns of H3K27me3 enrichment in chimpanzee and human iPSCs. Overlap across the two species is considerably higher than expected by chance (*Figure 5A*, $\chi^2$ p < 10$^{-16}$), but it is somewhat unclear how to interpret this observation with respect to the expectation that human and chimpanzee iPSCs would have similar pluripotency potential. We thus focused on a set of 3913 genes (*Li et al., 2013*) previously annotated as bivalently modified in human PSCs—that is, genes known to be associated with both high H3K4me3 and H3K27me3, indicative of a 'poised' or 'primed' state (*Bernstein et al., 2006*). We expect the vast

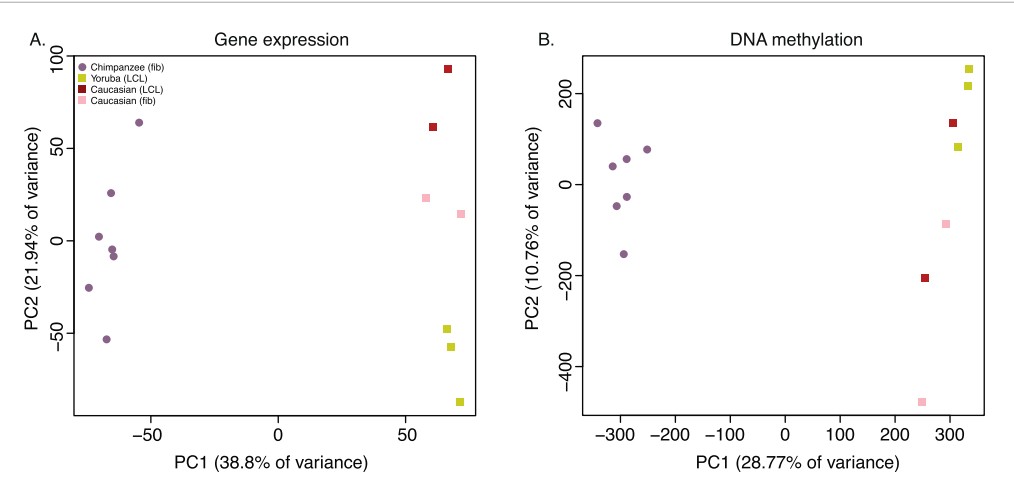

**Figure 4**. Principal component (PC) analysis plots of data from the iPSCs. (**A**) Principal component analysis (PCA) generated from expression data of 12,171 orthologous genes. (**B**) PCA generated from DNA methylation data measured by 335,307 filtered probes.

The following figure supplements are available for figure 4:

**Figure supplement 1**. Volcano plot showing the distribution of DE genes between iPSCs of chimpanzee and human origin.

**Figure supplement 2**. Density plots of log₂ FC change values amongst DE genes for the main comparisons presented in the text.

majority of these genes to also be associated with similar modifications in chimpanzee iPSCs. Only 2910 of the known bivalent genes were associated with clear orthologous TSSs and could be tested using our comparative H3K27me3 ChIP-seq data. Of these, 306 were not associated with the modification in either species, whereas of the 2604 genes that were associated with H3K27me3 in at least one species, 2368 (90.1%) were enriched for H3K27me3 in both species (*Figure 5B*, $\chi^2$ p < 10$^{-16}$).

We then examined H3K27ac enrichment patterns in both species. This mark is indicative of active promoters and gene transcription. Overall, we find good agreement between human and chimpanzee genes enriched for H3K27ac, with 95.8% human genes associated with the mark also enriched in chimpanzees (*Figure 5C*). However, there is a clear excess of genome-wide H3K27ac signal in chimpanzee iPSCs relative to humans, possibly due to an overall more sensitive ChIP enrichment in the chimpanzee samples (*Figure 5—figure supplements 1, 2*).

We proceeded by focusing on a list of 22 core pluripotency transcription factors (taken from *Ng and Surani, 2011*; *Orkin and Hochedlinger, 2011*), where we expect to find H3K27ac signal shared across the two species at a higher rate than in the genome-wide data, given the role of these factors in maintaining pluripotency. Due to our stringent requirements for establishing orthology, we were initially able to examine data from 14 of those genes; 11 of which were associated with H3K27ac in both species (*Figure 5D*)—one of the discrepancies is *REX1* (also known as *ZFP42*), which we discuss further below. We extended our analysis to include the full set of 22 pluripotency transcription factors regardless of orthology, by testing solely for absence or presence of signal peaks identified by MACS (i.e., without considering enrichment scores; see 'Materials and methods'). We again found a high overlap in H3K27ac enrichment across species, with 15 of the 22 genes associated with H3K27ac enrichment in both species (including the three master regulators of pluripotency, *OCT4*, *SOX2*, and *NANOG*; *Figure 5—figure supplement 3*). Of the remaining 7 genes, one (*DAX1*) was not found to be associated with H3K27ac in either species, four genes (*ESSRB*, *KLF2*, *KLF4*, and *KLF5*) were associated with H3K27ac only in chimpanzee (although this observation may reflect incomplete power to detect peaks in the human data), and only two genes (*ZFX* and *REX1*) were associated with H3K27ac in human but not in chimpanzee iPSCs.

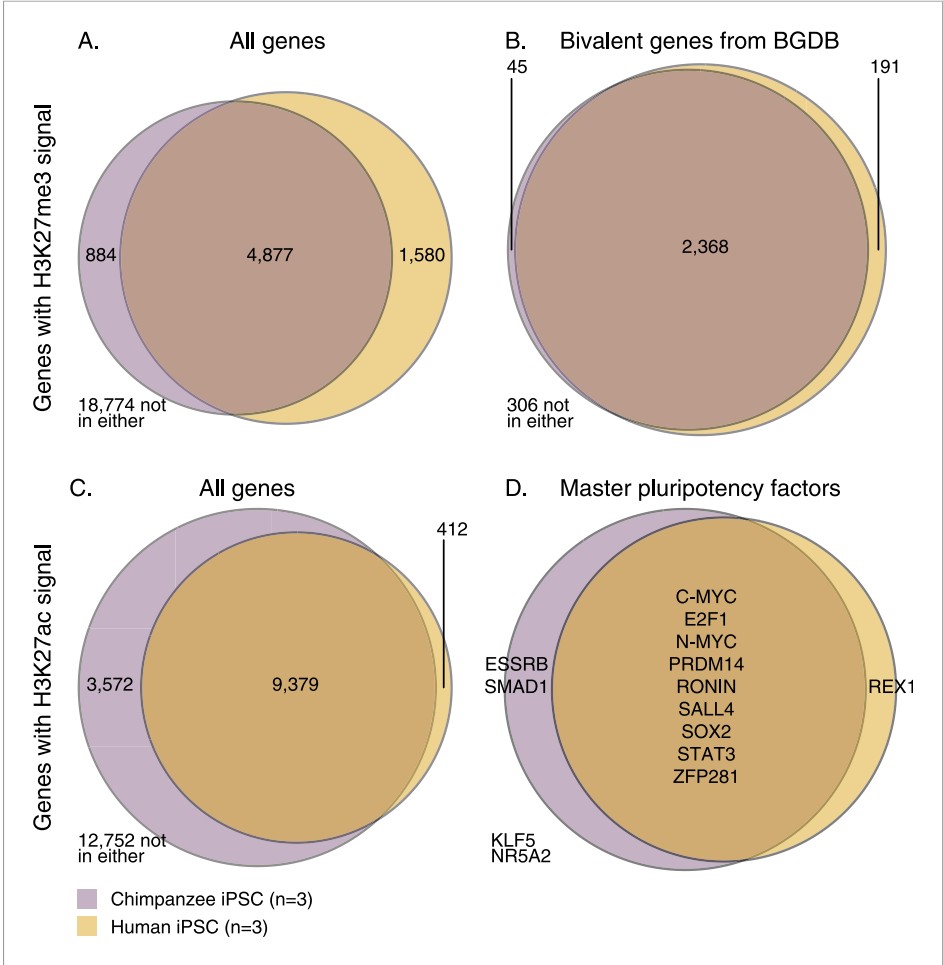

**Figure 5**. Overlap of H3K27me3 and H3K27ac signal between chimpanzee and human iPSCs at orthologous transcription start sites (TSSs). (**A**) H3K27me3 enrichment near all genes with an orthologous TSS. (**B**) H3K27me3 enrichment near 2910 genes previously identified as bivalent in human PSCs. (**C**) H3K27ac enrichment near all genes with an orthologous TSS. (**D**) H3K27ac peaks near 14 known pluripotency master regulators with orthologous TSSs.

The following figure supplements are available for figure 5:

**Figure supplement 1**. Density plots of H3K27ac enrichment scores at orthologous TSSs in the entire data set and at 3572 genes enriched only in chimpanzee iPSCs.

**Figure supplement 2**. Density plots of mean RPKM in chimpanzee iPSCs in all 12,171 genes with expression data and in the subset of 1737 genes with expression data and H3K27ac signal enrichment solely in chimpanzee iPSCs.

**Figure supplement 3**. H3K27ac peaks observed in at least 1 chimpanzee or human iPSC, as identified by MACS at 22 known pluripotency master regulators.

## REX1 may be dispensable for chimpanzee pluripotency

In order to further consider inter-species differences in the core pluripotency regulatory network, we examined expression levels in our chimpanzee and human iPSCs in the same list of 22 core pluripotency TFs described above. Expression values in all iPSC lines are shown in *Figure 6A* (see also *Figure 6—figure supplement 1*). Given the stringency of our interspecies analysis approach with respect to unique read mapping, we are unable to calculate RNA-seq-based expression estimates for six of these TFs, including *OCT4* or *NANOG*, both of which have multiple pseudogenes that can confound mapping algorithms (however, as shown in *Figure 1D*, our qPCR results demonstrate that

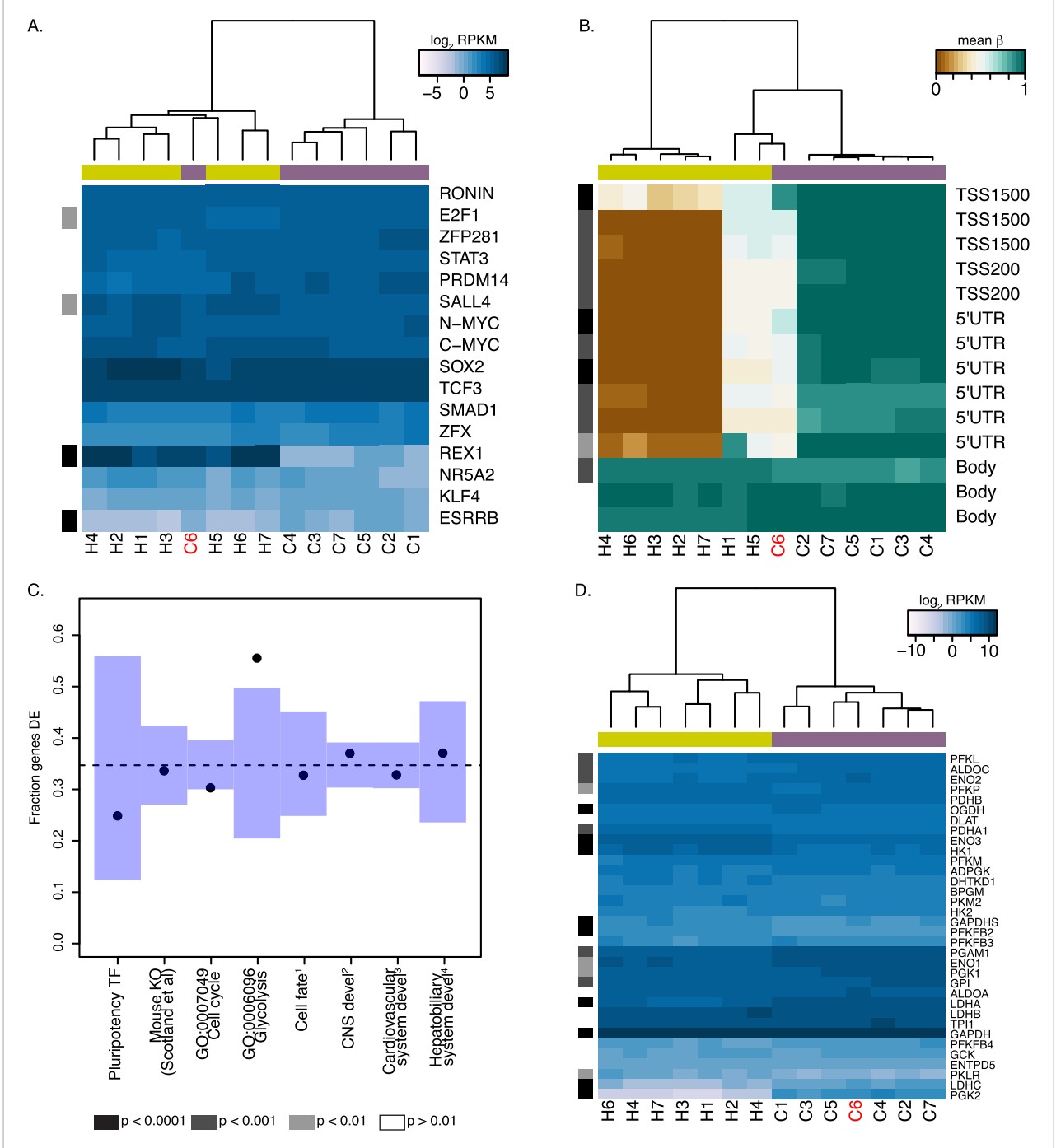

**Figure 6**. *REX1* may be dispensable for pluripotency in chimpanzee iPSCs. In both panels REX1-expressing chimpanzee iPSC line is coloured red, significant interspecies differences are indicated along the left-hand side, and purple boxes indicate chimpanzee lines, yellow boxes indicate human lines. (**A**) Expression values of 16 core pluripotency transcription factors in all human and chimpanzee iPSC lines. (**B**) Methylation status of 13 CpG sites associated with *REX1* in all human and chimpanzee iPSCs. Location of the probe relative to the gene sequence is indicated along the right hand side. (**C**) Fraction of differentially expressed (DE) genes in multiple categories downstream of *REX1* in human and mouse ESCs. 1: Genes associated with any Gene Ontology term that contains the words 'ectoderm', 'mesoderm' or 'endoderm'. 2: CNS development genes are associated with GO:0007417 or any of its offspring. 3: cardiovascular system development genes are associated with GO:0072358 or any of its offspring. 4: hepatobiliary system development genes are associated with GO:0055123 or any of its offspring. (**D**) Expression levels of 34 genes associated with GO:0006096, glycolysis, in all human and chimpanzee iPSC lines. All reported p-values were calculated after excluding C6.

*Figure 6. continued on next page*

*Figure 6. Continued*

The following figure supplements are available for figure 6:

**Figure supplement 1**. Expression values of 15 core pluripotency transcription factors in all human and chimpanzee iPSC lines.

**Figure supplement 2**. Expression levels of *REX1* in human, chimpanzee and bonobo iPSC lines generated in this study and in *Marchetto et al. (2013b)*.

**Figure supplement 3**. Plot of PluriTest pluripotency scores vs normalised *REX1* intensity in 73 human iPSC lines derived in-house.

**Figure supplement 4**. Methylation status of 13 CpG sites associated with *REX1* in chimpanzee and human iPSCs from this study and human PSCs from *Ziller et al. (2011)*.

expression of those 2 genes is similar amongst all chimpanzee iPSC lines, and marginally higher than in our human iPSC control line). Of the 16 TFs with expression data for iPSCs from both species, 4 (*E2F1*, *ESRRB*, *SALL4* and *REX1*) are DE between human and chimpanzee iPSCs at an FDR of 1%. Of these, *ESRRB* and *REX1* are associated with absolute inter-species expression $\log_2$ fold-changes >1. However, because *ESRRB* is expressed at very low levels across all samples (mean RPKM across all 14 samples = 0.47), we focused our subsequent analyses on *REX1*, which is expressed at low or undetectable levels in 6 of our 7 chimpanzee iPSCs (mean RPKM = 0.667), but at high levels in all human iPSC lines (mean RPKM = 180.58) and a single chimpanzee iPSC, C6 (*Figure 6A*). Our DNA methylation data is consistent with this gene expression pattern: all 10 probes located in the 5′ UTR or up to 1500 bp upstream from the *REX1* TSS are highly methylated in the six chimpanzee lines (mean β across all promoter probes = 0.87), but exhibit intermediate or low levels of DNA methylation in all of the human iPSC lines and the *REX1*-expressing C6 line (*Figure 6B*); the entire region is a DMR (*Supplementary file 5*). Consistent with these findings, *REX1* is also differentially enriched for H3K27ac signal in the two species—we identified no H3K27ac peaks at the *REX1* TSS in the three chimpanzee lines, which did not include C6 (*Figure 5D*, *Figure 5—figure supplement 3*).

The *REX1* genes codes for a transcription factor present in all placental mammal species, which has long been established as a marker of pluripotency in human and mouse PSCs (*Brivanlou et al., 2003*). Multiple publications have suggested that this gene plays an important role in maintaining pluripotency and inhibiting differentiation into the three primary tissue germ layers (*Masui et al., 2008*; *Scotland et al., 2009*; *Son et al., 2013*), with multiple mechanisms of action having been proposed. However, *REX1*-knockout mouse ESC lines can give rise to chimeric animals, and homozygous F2 *REX1* null mice are viable (*Masui et al., 2008*), suggesting that *REX1* may not be indispensable for murine pluripotency. In humans, loss of *REX1* expression in ESCs following shRNA knockdown has been associated with a rapid loss of pluripotency, as well as a decrease in glycolytic activity and a lack of observable mature mesodermal structures in teratoma formation assays (*Son et al., 2013*).

To determine the consequences of a lack of *REX1* expression in chimpanzee iPSCs, we considered gene expression data from all human iPSC lines and the 6 chimpanzee iPSC lines that do not express *REX1*. We asked whether there is an excess of DE genes among those thought to be directly regulated by, or downstream of, *REX1* (*Figure 6C,D*; see 'Materials and methods'), but failed to find enrichment in all categories except for genes associated with GO term BP:0006096, glycolysis, where 19 of 34 testable genes were DE at an FDR of 1% between the two species (p < 0.01 from 100,000 permutations). The direction of this effect ran contrary to previous reports, however, with genes highlighted by *Son et al. (2013)* as downregulated following *REX1* knockdown, such as *PGAM1* or *LDHA*, having significantly higher expression in chimpanzee iPSCs than in human iPSCS (*Figure 6D*). Furthermore, the *REX1*-expressing line C6 is not an outlier amongst the other chimpanzee iPSC lines (*Figure 6D*), suggesting that the observed inter-species regulatory differences cannot be attributed to differences in *REX1* expression between the species.

We note that both the teratomas and EBs generated from chimpanzee iPSC lines that do not express *REX1* gave rise to mature structures from all three germ layers similar to those observed in *REX1*-expressing line C6 (*Figure 2—figure supplements 1, 3*). Furthermore, and consistent with our observations, *REX1* is either absent or expressed at low levels in one replicate of either of the two

retrovirally reprogrammed bonobo (*Pan paniscus*, sister species to chimpanzees) iPSC lines generated by *Marchetto et al. (2013b)*, although it is expressed in both replicates of both chimpanzee iPCS lines from the same group (*Figure 6—figure supplement 2*). Together, these findings suggest that that the variable loss of *REX1* expression in chimpanzee and bonobo iPSCs does not impair pluripotency, and that its regulatory functions of in humans may be being fulfilled in chimpanzee iPSCs by other regulatory mechanisms.

## Comparison of iPSCs and other tissues

We collected RNA-sequencing data from all cell lines used to generate both the chimpanzee and human iPSCs (*Supplementary file 6*). Following quality control and normalisation steps, we obtained RPKM values for 13,147 genes across all 28 iPSC and precursor samples (see 'Materials and methods'). We also obtained DNA methylation profiles from all samples at the same 335,307 probes described above. PCA of both data sets show that the first PC was significantly associated with tissue type in both data sets ($p < 10^{-27}$ for the expression data; $p < 10^{-17}$ for the DNA methylation data; see *Figure 7* and *Supplementary file 9*), while human and chimpanzee samples are separated by species along PC2 ($p = 0.001$ for the expression data; $p < 10^{-4}$ for the methylation data). However, given the absence of chimpanzee LCLs in our dataset, it is not possible to determine whether the separation is driven by tissue type, species, or both.

Overall, chimpanzee iPSCs have significantly higher levels of DNA methylation compared to the somatic lines they were generated from ($p < 10^{-15}$; *Figure 7—figure supplement 1*), an observation that extends to all genomic features we tested (*Figure 7—figure supplement 2*); similar observations have been previously made in human PSCs (*Bock et al., 2011*; *Nazor et al., 2012*). Remarkably, both DNA methylation and gene expression levels in iPSCs are relatively homogeneous within species, far more so than in their corresponding precursor cells (*Figure 6B,D*; $p < 10^{-14}$ when comparing overall pairwise distances within all chimpanzee iPSCs and within all chimpanzee fibroblasts in the methylation data; $p < 10^{-9}$ for the same comparison in the gene expression data). DNA methylation levels in iPSCs also have significantly reduced coefficients of variation relative to their precursor lines (range of CVs for chimpanzee iPSCs = 0.78–0.80, for chimpanzee fibroblasts = 0.87–0.90; $p < 10^{-06}$). We observed the same pattern in the human data, although in this case the multiple somatic origins of the cell lines of origin contribute to the higher level of variation.

We then performed analyses of gene expression and DNA methylation differences in the combined iPSC and somatic precursor dataset. First, we carried out a comparison of the iPSCs and the precursor cells within each species (see 'Materials and methods') and classified 9235 genes as DE between chimpanzee fibroblasts and the corresponding iPSCs. In humans the number of DE genes is 7765 if we consider all iPSC lines and their somatic precursors, 8087 if we only consider those derived from LCLs (n = 5), and 5489 if we only consider those derived from fibroblasts (n = 2; *Supplementary file 10*). Similarly, we identified 18,029 DMRs between chimpanzee fibroblasts and iPSCs, and 12,078 DMRs between all human somatic precursors and all human iPSCs (*Supplementary files 11, 12*). No GO categories are significantly overrepresented in any of these data sets.

Next, we focused on a comparison of inter-species differences in gene expression and DNA methylation levels across cell types. Following joint normalisation and modelling of data from all samples (see 'Materials and methods'), we classified 5663 genes as DE between the chimpanzee precursor fibroblasts and the collection of human precursor LCLs and fibroblasts, as well as 84,747 DM probes and 9107 DMRs (always at an FDR of 1%). Most of these regulatory differences, however, reflect variation across cell types rather than across species (6324 genes and 70,312 probes are DE or DM between the human fibroblasts and LCLs, respectively). We thus considered only data from the fibroblast precursors in the two species. Only 2 of the human iPSCs were reprogrammed from fibroblasts, leading to a loss in power; we were nonetheless able to identify 1236 DE genes and 25,456 DM probes between human and chimpanzee fibroblasts, and 1118 DE genes and 16,392 DM probes between the corresponding iPSCs of the two species. None of these gene sets were significantly enriched for functional annotations using GO BP terms. Although the overlap of inter-species DE genes and DM probes between the iPSCs and the precursors is considerable (13.6% of DE genes and 11.8% of DM probes), a large number of regulatory differences are only observed between the iPSC lines of the two species (*Figure 7—figure supplement 3*). This observation is robust with respect to different approaches to normalising and modelling the data (*Figure 7—figure supplement 4*),

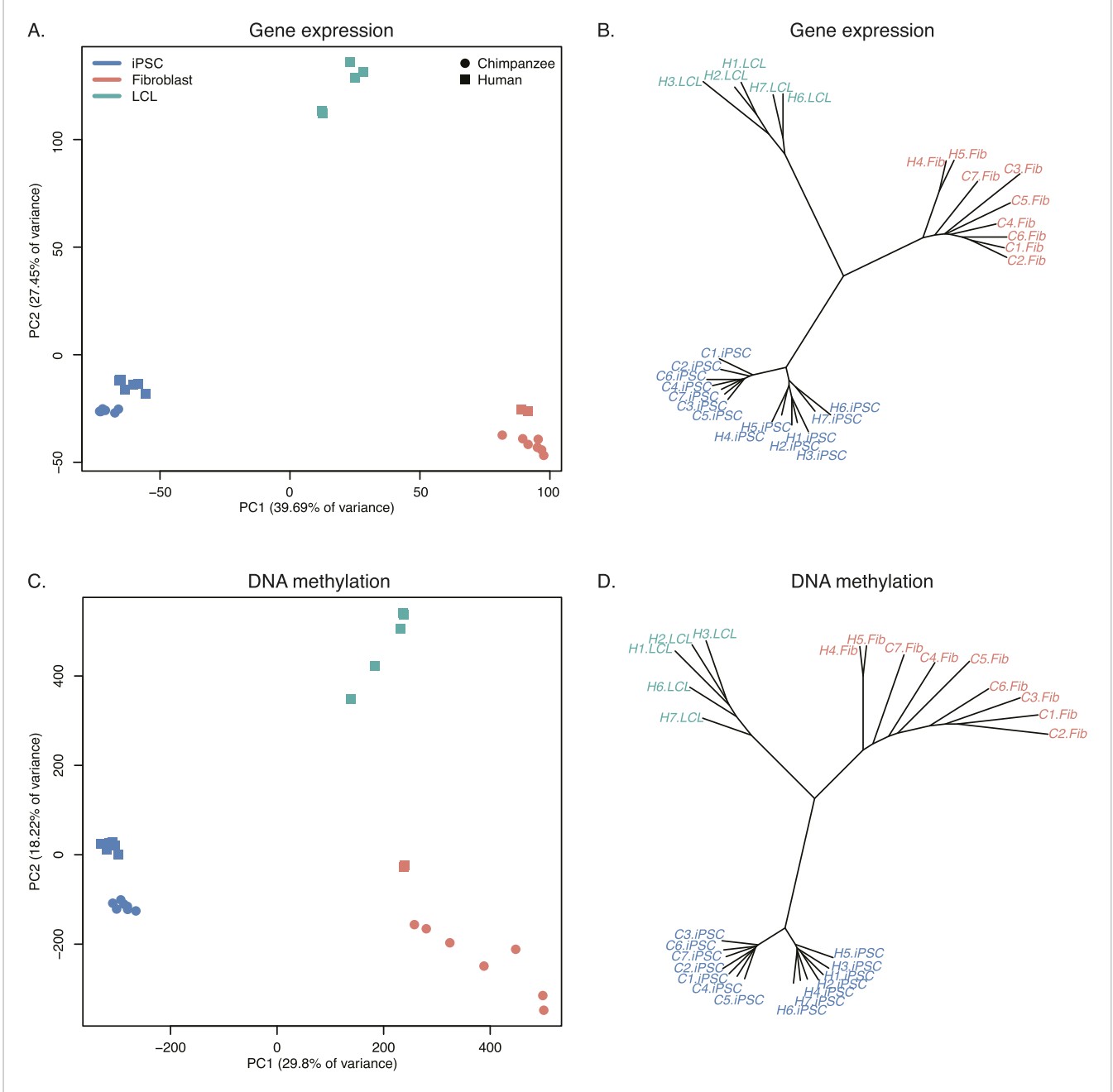

**Figure 7**. Relationships of iPSCs to their precursors. (**A**) PCA of gene expression data from all iPSCs and their precursor cell lines. (**B**) Neighbour-joining tree of Euclidean distances between all samples generated based on the gene expression data. (**C**) PCA of DNA methylation data from all iPSCs and their precursor cell lines. (**D**) Neighbour-joining tree of Euclidean distances between all samples generated based on the DNA methylation data.

The following figure supplements are available for figure 7:

**Figure supplement 1**. Boxplots of methylation beta values at 335,307 probes across all samples.

**Figure supplement 2**. Boxplots of methylation beta values across all samples, grouped by potency and genomic features.

**Figure supplement 3**. Venn diagrams showing overlap in interspecies differences before and after reprogramming.

*Figure 7. continued on next page*

*Figure 7. Continued*

**Figure supplement 4**. Venn diagram showing overlap of genes identified as DE between iPSCs of the two species when we normalize the iPSC data independently and alongside data from the precursors.

**Figure supplement 5**. Venn diagram showing overlap of probes identified as DM between iPSCs of the two species under the full and reduced *limma* models.

**Figure supplement 6**. Normalized *XIST* expression values in 7 chimpanzee and human iPSCs.

**Figure supplement 7**. Quantile-normalized methylation beta values at 8210 X-chromosome probes in 7 chimpanzee iPSCs and 7 human iPSCs.

**Figure supplement 8**. Normalized methylation beta values at 168 assayable probes known to be subject to parental imprinting effects, from *Ma et al. (2014)*.

strongly suggesting that many of the differences we observe between our chimpanzee and human iPSC lines may be intrinsic features of the pluripotent state in these two species.

## Discussion

iPSCs have the potential to transform our understanding of the biology of non-model organisms and facilitate functional comparative studies. To this end, we have generated a panel of 7 fully characterized chimpanzee iPSCs. All lines are capable of spontaneously giving rise to the three tissue germ layers in vitro and in vivo and meet all currently established criteria for pluripotency. The chimpanzee iPSC lines provide a tantalising avenue for investigating how changes in gene expression and regulation underlie the architecture of complex phenotypic traits in humans and our closest living relatives (*Gallego Romero et al., 2012*; *Marchetto et al., 2013a*). In particular, we believe that through the use of directed differentiation protocols, functional studies could be performed in cell types where strong a priori hypotheses support a role for selective pressure underlying inter-species divergence (e.g., liver, heart, kidney [*Blekhman et al., 2008*, *2010*]). In that sense, we hope that this panel of cell lines will be a useful tool to researchers interested in overcoming current limitations of comparative studies in primates. To that purpose, all chimpanzee iPSC described in this publication the panel are available fully and without restrictions to other investigators upon request to the corresponding authors.

Other groups have previously generated pluripotent stem cells from primates (*Liu et al., 2008*; *Chan et al., 2010*; *Tomioka et al., 2010*; *Wu et al., 2010*; *Ben-Nun et al., 2011*; *Deleidi et al., 2011*; *Okamoto and Takahashi, 2011*; *Wu et al., 2012*; *Hong et al., 2014*; *Wunderlich et al., 2014*). Indeed, a recent publication (*Marchetto et al., 2013b*) reported the generation of two chimpanzee and two bonobo (*P. paniscus*) iPSC lines through the use of retroviral vectors. However, in the course of our work we have found that retroviral vector silencing in chimpanzee iPSCs was not as stable as in human iPSC lines generated at the same time using the same method (see 'Materials and methods' and *Figure 1—figure supplement 5*). Our use of episomal vectors circumvents this problem, and more broadly the problems of both random exogenous gene reactivation and disruption of the host genome through retroviral integration (*Sommer et al., 2012*).

More generally, while the sum total of primate PSC generation efforts so far has resulted in a sizable number of lines being established from various donors and species, these have been generated through various reprogramming protocols and source cell types. We have generated iPSCs from a panel of seven individuals using a consistent protocol and cell type of origin. Given the panel size, it is powerful enough to robustly detect inter-species differences in gene expression, splicing and regulation. The fact that our panel contains both female and male lines also allows for future studies of sex-specific differences in gene expression in various cell types. Indeed, we have previously shown that this can be accomplished using as few as six individuals from each species (*Blekhman et al., 2010*).

Beyond its future applications, however, our panel has already yielded insights into the pluripotent state in chimpanzees and humans. On the one hand, both at the transcriptional and epigenetic level, our iPSCs are remarkably homogeneous both within and between species, significantly more so than their precursors cells. This finding aligns with our current understanding of the reprogrammed

pluripotent state as a complex, highly regulated state (*Jaenisch and Young, 2008*), deviations from which are likely to result in loss of pluripotency and lineage commitment. Additional support for this notion was provided by the strong overlap in H3K27me3 signal between the two species, especially in known bivalent genes. It is remarkable that we have been able to observe this considerable conserved chromatin signature despite the obvious confounding technical batch effect in these comparative data.

On the other hand, we were also able to identify over 4500 genes that are DE between human and chimpanzee iPSCs, as well as over 3500 DMRs between the two species. These numbers are greater than what has been previously observed in comparisons of other tissues across humans and chimpanzees with similar sample sizes (*Blekhman et al., 2008*, *2010*). We believe that the reasons for this difference are likely to primarily stem from increased power to detect DE genes and DMRs in our sample relative to previous work. Given the small amount of intra-species variability we observed in RNA-seq and methylation relative to other tissues, we expect to have greater power to detect small, statistically significantly inter-species differences that would have been missed by studies that consider more variable tissue samples. This notion is supported by the fact that the fraction of genes with log FC < 2 we detect as DE between human and chimpanzee iPSCs is greater than in other comparison we have performed with any other tissue (*Figure 4—figure supplement 2*). Though small in magnitude, we expect that a subset of these regulatory differences may be biologically relevant (e.g., we find that inter-species regulatory differences in methylation levels are enriched in regions associated with developmental processes; *Supplementary files 4, 5*).

We specifically highlighted an inter-species difference in *REX1* expression levels. This gene is considered an indispensible pluripotency marker in human and mouse PSCs, but our observations suggest that it may not be the case in chimpanzees. Although only one chimpanzee iPSC line expresses *REX1*, we were unable to identify any systematic differences between our human and chimpanzee iPSCs that would indicate a reduction in pluripotency. We also examined *REX1* expression levels in 73 human iPSC lines generated in-house from Caucasian individuals using the Illumina HT12v4 array (*Figure 6—figure supplement 3*). All lines had PluriTest pluripotency scores >20, yet 3 of 73 lines (4.1%) showed levels of *REX1* expression that were indistinguishable from background signal, suggesting that *REX1* may not be expressed in these individuals despite their high pluripotency scores. We also examined methylation status at the *REX1* locus in previously published human ESCs and iPSCs from *Ziller et al. (2011)*, and found that although all ESC lines examined exhibited consistent levels of low methylation at the *REX1* promoter, human iPSC lines analysed in exhibited either hemi- or hyper-methylated *REX1* promoter regions (*Figure 6—figure supplement 4*). In the absence of publicly available *REX1* expression data from either of the hiPSC lines with hypermethylated promoters we cannot be certain that the gene is not expressed in these lines, but the combination of these findings with our observations above and previous literature suggest that *REX1* may be important in regulation and maintenance of pluripotency in ESCs, but not necessarily so in iPSCS.

Additionally, in chimpanzees, the *REX1* gene has undergone multiple deletions and insertions relative to the human sequence, most significantly a 647 bp insertion in its first intron, and two insertions in the 3′ UTR region of approximately 300 bp each that may disrupt the local regulatory landscape; the gene has also been duplicated, with a second copy retrotransposed into chromosome 14; none of these changes are shared with gorillas or orang-utans. Although it is currently unclear whether some or all of these changes are also present in the bonobo, these findings might explain why we observed low or no *REX1* expression more frequently in chimpanzees than in humans, and suggests that the gene may not be necessary for maintaining pluripotency in the *Pan* lineage.

PSCs have been used to study developmental pathways in vitro (e.g., *Paige et al., 2012*; *Rada-Iglesias et al., 2012*; *Wamstad et al., 2012*; *Xie et al., 2013*). Although optimization of existing differentiation protocols will likely be necessary for application in the chimpanzee system, our panel of iPSCs makes it possible to carry out comparative developmental studies between humans and chimpanzees, and firmly test the hypothesis that changes in gene regulation and expression, especially during development, underlie phenotypic differences between closely related species, especially primates (*Britten and Davidson, 1971*; *King and Wilson, 1975*; *Jacob, 1977*; *Carroll, 2005*, *2008*). In addition, we should be able to recreate and test the effect of inter-species regulatory changes in the correct cell type and species environment, enabling studies that cannot otherwise be performed in humans and non-human primates. The use of panels of iPSCs including lines from both humans and non-human primates will thus allow us to gain unique insights into the genetic and regulatory basis for human-specific adaptations.

## Materials and methods

### Isolation and culture of fibroblasts

All biopsies and animal care were conducted by the Yerkes Primate Research Center of Emory University under protocol 006–12, in full accordance with IACUC protocols. Skin punch biopsies (3 mm) were rinsed in DPBS containing Primocin (InvivoGen, San Diego, California) and penicillin/streptomycin (Pen/Strep, Corning, Corning, New York) and manually dissected into 10–15 smaller pieces. The tissue was digested in 0.5% collagenase B (Roche, Indianapolis, Indiana) for 1–2 hr until cells were released from the extracellular matrix. Dissociated cells were pelleted by centrifugation at $250 \times g$, and the supernatant was spun a second time at $700 \times g$ to pellet any cells that had not been completely released from the extracellular matrix. Cell pellets were resuspended in a 1:1 mixture of α-MEM and F12 (both from Life Technologies, Carlsbad, California) supplemented with 10% FBS (JR Scientific, Woodlawn, California), NEAA, GlutaMAX (both from Life Technologies), 1% Pen/Strep, 64 mg/l L-ascorbic acid 2-phosphate sesquimagnesium salt hydrate (Santa Cruz Biotech, Dallas, Texas) and Primocin. Cells were plated in a single well of a 6-well plate coated with 4 μg/cm$^2$ human fibronectin (BD Sciences, Franklin Lakes, New Jersey) and 2 μg/cm$^2$ mouse laminin (Stemgent, San Diego, California). Cultures were grown at 5% $CO_2$/5% $O_2$ until confluent and then split using 0.05% trypsin. For routine passaging cells were cultured at 5% $CO_2$ and atmospheric oxygen in primate fibroblast media, which is the same as plating media but does not contain F12 base media.

### Generation of retrovirally-reprogrammed iPSC lines (a failed attempt)

We initially attempted to generate lines by retroviral transduction through transfection with pMXs-vectors encoding the human *OCT3/4*, *SOX2*, *KLF4*, *L-MYC* and *NANOG* sequences (Addgene plasmids 17,217, 17,218, 17,219, 26,022 and 18,115) as well as vectors encoding the MSCV-VSV.G envelope protein (Addgene plasmid 14,888) and MSCV gag-pol (Addgene plasmid 14,887). 15 μg of each vector was transfected into 293FT cells (Life Technologies) using Lipofectamine 2000 (Life Technologies) as directed by the manufacturer. We collected virus-containing supernatant from the 293FT cells 48 and 72 hr after transfection and immediately used this viral media to transduce chimpanzee fibroblasts, alongside 10 μg/ml of polybrene (H9268 from Sigma Aldrich [St Louis, Missouri]). To aid viral penetration, we centrifuged the cells at 1800 RPM for 45 min following each transduction. 24 hr after the second transduction, we replaced the viral media with A-MEM + 10% FBS, NEAA and Glutamax. Transduced fibroblasts were allowed to recover for a further 2 days and then seeded on γ-irradiated, CF-1-derived MEF at a density of 10,000 cells/cm$^2$, and maintained in hESC media (DMEM/F12 supplemented with 20% KOSR, 0.1 mM NEAA, 2 mM GlutaMAX, 1% Pen/Strep, 0.1 mM BME and 25 ng/ml human bFGF) supplemented with 0.5 mM valproic acid (Stemgent) until day 14. We obtained iPSCs from 5 chimpanzees by using this protocol. Yet, when we performed quality control and pluripotency checks on these lines we found that the exogenous transfected genes were still expressed (*Figure 1—figure supplement 5*). Pluripotency in these lines could not be maintained exclusively through endogenous expression. We discarded all 5 lines and proceeded with a different reprograming strategy as detailed below.

### Generation of episomally-reprogrammed iPSC lines

Fibroblasts were grown at 5% $CO_2$/atmospheric $O_2$ in primate fibroblast media until 70–80% confluence and released by trypsinisation for transfection. $1.5 \times 10^6$ cells were transfected with 1.5 μg per episomal vector containing the following genes: *OCT3/4, SHp53, SOX2, KLF4, LIN28*, and L-MYC (Addgene plasmids 27,077, 27,078, 27,080 and 27,082; [*Okita et al., 2011*]). To boost the initial retention of vectors following transfection, 3 μg of in vitro transcribed ARCA capped/polyadenylated EBNA1 mRNA was cotransfected with the vectors (see below). Transfected cells were seeded at 15,000/cm$^2$ on tissue culture plates precoated with 1 μg/cm$^2$ vitronectin (Stemcell Technologies, Vancouver, Canada). Cells were grown in Essential 8 media (made in house as previously described in *Chen et al. (2011)*) without *TGFβ1*, supplemented with 0.5 mM sodium butyrate (NaB, Stemgent) and 100 nM hydrocortisone (Sigma Aldritch). Hydrocortisone was used between days 1–12, or until cell density exceeded >70% confluence. At day 12, cells were detached using TrypLE (Life Technologies) and replated at a density of 5000 cells/cm$^2$ on cell culture dishes precoated with 0.01 mg/cm$^2$ (1:100) of hESC-grade Matrigel (BD Sciences) and grow in Essential 8 media without *TGFβ1* or NaB. Colonies

began to form at days 18–22 and were picked between days 24–30 onto dishes coated with γ-irradiated CF-1 derived MEF and subsequently grown in hESC media (as described above) supplemented with 100 ng/ml human bFGF (Miltenyi Biotech, Teterow, Germany). Clones were routinely split using Rho-associated kinase (ROCK) inhibitor Y27632 (Tocris, Minneapolis, Minnesota) at a concentration of 10 µM. Cells were migrated to 1:100 hESC Matrigel (BD Sciences) and maintained on Essential 8 media after a minimum of 15 passages on MEF. Feeder free cells were passaged using EDTA-based cell release solution as in Chen et al. (2011).

## Generation of *EBNA1* mRNA

To generate a template for in vitro transcription, an EBNA1 template was designed using the wild type HHV4 *EBNA1* as a reference sequence (NCBI accession YP_401677.1). The reference sequence was modified by replacing the GA repeat region and domain B (amino acids 90–375) with a second, tandem, chromatin-binding domain (domain A, amino acids 27–89), similar to what was done by Howden et al. (2006). The nuclear localization signal (amino acids 379–386) was removed and replaced with the sequence GRSS. Using the amino acid sequence as the starting template, the corresponding DNA sequence was generated by reverse translation and optimized for expression in human cell lines using Genscript's OptimumGene codon algorithm. This sequence was synthesized by Genscript (Piscataway, New Jersey) and provided in the pUC57 cloning vector; the EBNA1 coding sequence was subcloned into pcDNA3.1+ (Life Technologies) using the restriction enzymes BamHI and HindIII. Capped and poly(A) mRNA transcripts were generated using the mMESSAGE mMACHINE T7 ULTRA kit (Life Technologies) with 1 µg of BamHI linearized pcDNA3.1+EBNA1 as the template. The plasmids encoding the wild type and modified EBNA1 sequences have been deposited to Addgene as plasmid ID#s 59,199 and 59,198 for the wild type and modified sequences respectively.

## iPSC characterization by immunocytochemistry

iPSC colonies were cultured on MEF for 4–6 days and fixed using PBS containing 4% PFA (Santa Cruz Biotech) for 15 min at room temperature. After rinsing with PBS, fixed cells were blocked and permeabilised for 1 hr in PBS containing 0.3% triton and 5% BSA. Primary antibodies: OCT3/4 (SC-5279), SOX2 (SC-17320), NANOG (SC-33759), SSEA-4 (SC-21704), and Tra-1-81 (SC-21706), all from Santa Cruz Biotech, were diluted 1:100 in blocking solution. Fixed cells were incubated with the primary antibody solution overnight on a rocker at 4°C. After washing out the primary antibody solution, fixed cells were incubated with secondary antibodies (labeled with either Alexa-488 or Alexa-594, 1:400, Life Technologies) diluted in blocking for 1 hr on a rocker at room temperature. Nuclei were counterstained using 1 µg/ml Hoechst 33,342 (ThermoFisher Scientific, Waltham, Massachussets). All fluorescence imaging was conducted using an AMG EVOS FL (Life Technologies).

## Quantitative PCR for endogenous and exogenous gene expression

RNA was extracted using Qiagen RNA miniprep columns from cell pellets collected from fibroblasts, day 7 post transfection and feeder free (Matrigel and Essential 8) iPSC lines at passage 10 or higher for both the retroviral and episomal reprogrammings; 1 µg of total RNA was reverse transcribed using the Maxima first strand cDNA synthesis kit (ThermoFisher Scientific). Quantitative PCR was performed using a 1:96 dilution of cDNA and SYBR Select master mix (Life Technologies) with both forward and reverse primers at a concentration of 0.2 µM. Data was collected and analysed using the Viia7 (Life Technologies). Primer sequences are shown in *Supplementary file 2*, exogenous gene expression melt curves are shown in *Figure 1—figure supplement 5*.

## Generation of embryoid bodies and immunofluorescence

Colonies growing on MEF were detached using Dispase/Collagenase IV (1 mg/ml each; both from Life Technologies) in DMEM/F12 and grown as a suspension culture on low adherent plates using hESC media without bFGF. After 1 week of suspension growth, cells were transferred to 12 or 24-well plates coated with 0.1% gelatin and grown in DMEM supplemented with 20% FBS, 0.1 mM nonessential amino acids, 2 mM GlutaMAX, 1% Pen/Strep and 64 µg/ml L-Ascorbic acid 2-phosphate sesquimagnesium salt hydrate. Embryoid bodies were grown for 1–2 weeks prior to fixation and immunofluorescence staining. Cultures were fixed and stained as described above using the following antibodies: AFP (1:200, SC-130302, Santa Cruz Biotech), FOXA2 (1:200, SC-6554, Santa Cruz Biotech), α-smooth muscle actin (1:

1500, CBL171, EMD Millipore, Billerica, Massachussets) and MAP2 (1:200, sc-20172 and sc-74420, Santa Cruz Biotech).

## Integration analysis

To test for genomic integration and residual retention of episomal plasmids, each iPSC line was migrated to feeder free conditions and grown beyond passage 15 on hESC-qualified Matrigel (1:100 dilution, BD Sciences) coated plates in Essential 8 media (Life Technologies). DNA was extracted from feeder free cultures using DNeasy Blood and Tissue Kits (Qiagen, Valencia, California). PCR was performed using 100 ng of genomic DNA, an annealing temperature of 72°C and 25 cycles using primers designed to amplify a region common to all episomal vectors used (*Supplementary file 2*). Genomic DNA (100 ng) isolated from day 7 cultures, and 1 pg of each episomal vector were used as positive controls. PCR products were run on a 1% agarose gel and visualised using ethidium bromide.

## Karyotyping

After 15 passages on MEF and hESC media, cells were migrated to 1:100 hESC Matrigel (BD Sciences) and maintained on Essential 8 media for upwards of 6 passages. Feeder-free adapted cells were sent to Cell Line Genetics Inc (Madison, WI) for karyotyping as described in *Meisner and Johnson (2008)*.

## Teratoma formation assays

In vivo developmental potential of the reprogrammed cell lines was examined. Monolayer iPSCs from three chimpanzee lines were grown on Matrigel (1:100) in E8 medium (Life Technologies) and collected by EDTA treatment (Life Technologies). Cells were counted and resuspended at a ratio of 1:1 cell volume to Matrigel and kept on ice until the injection. 6-week-old CB17.Cg-*Prkdc$^{scid}$Lyst$^{bg-J}$*/Crl immunodeficient male mice were obtained (Charles River Laboratories, Wilmington, Massachussets) and approximately 1 million iPSCs for each clone were injected into the testis-capsule. After 5–8 weeks teratomas were isolated, weighed, measured, dissected, and fixed in 10% formalin. The specimens were embedded in paraffin, stained with hematoxylin and eosin, and analyzed by a histopathologist. All animal work was conducted under the approval of the Institutional Care and Use Committee of UCSD (Protocol# S09090).

In addition, live feeder free iPSC cultures maintained in Essential 8 media on Matrigel iPSCs from C4955 (passage 15 + 7) were provided to Applied Stem Cell Inc. (Menlo Park, CA) for teratoma analysis as previously described (*Chen et al., 2012*).

## Species-of-origin identity of teratoma samples

DNA was extracted from frozen teratoma tissue using DNeasy Blood and Tissue Kits (Qiagen). For teratomas derived from individual C4955, core sections were isolated from FFPE embedded teratomas tissue using a 3 mm dermal punch tool; DNA was extracted from core samples using a QIAamp DNA FFPE Tissue Kit (Qiagen). PCR was performed using universal mitochondrial primers ([*Kocher et al., 1989*] *Supplementary file 2*) amplifying cytochrome b (*Cytb*, chimpanzee reference sequence NC_001643:bp 14,233–14,598) or the 12S ribosomal gene (*12S*, NC_001643: bp 484–915) with 250–500 ng of genomic DNA as the starting template. Two-step PCR was conducted with an annealing temperature of 50°C for 1 min and an extension step at 72°C for 4 min for a total of 30 cycles. DNA was purified using a Wizard SV gel and PCR Clean-up kit (Promega, Madison, Wisconsin); dye terminator cycle sequencing was conducted by the University of Chicago Comprehensive Cancer Center using 60 ng of purified PCR template and 4 µM of either the forward or reverse primer. Alignment to the chimpanzee, human (NC_012920) and mouse (NC_005089) reference sequences was accomplished using CLC Main Workbench 6.9 (Qiagen) and MUSCLE (*Edgar, 2004*).

## Directed differentiation of chimpanzee iPSCs to hepatocytes and cardiomyocytes

In order to demonstrate that chimpanzee iPSCs can be directly differentiated into other cell types, we differentiated C2 iPSC into hepatocytes and C7 into cardiomyocytes using the published protocols of *Cheng et al. (2012)* and *Lian et al. (2013)* respectively, with the following modifications: In both cases we plated iPSCs at $0.35 \times 10^6$ cells/cm² in 0.44 ml/cm² and cultured them in Essential 8 media 24 hr prior to initiating all differentiations. To increase hepatocyte differentiation efficiency, 1 µM of sodium

butyrate was added during the first 24 hr of differentiation. After 24 days of differentiation, cells were immunostained as described above with a primary antibody for albumin (1:200, A6684, Sigma Aldrich; *Figure 2—figure supplement 2*).

After 10 days of differentiation, differentiated C7 cultures were enriched for cardiomyoctes by culture in RPMI based media without glucose supplemented with 5 mM sodium DL-lactate for 10 days as described previously (*Tohyama et al., 2013*; *Burridge et al., 2014*). After day 20 purified cardiomyocytes were cultured in media lacking glucose supplemented with 10 mM galactose (*Rana et al., 2012*). After 25 days of cardiac differentiation, we characterized calcium flux in and out of iPSC-derived cardiomyocytes by treating cultures with 5 μM Fluo-4 AM (F-14217, Life Technologies) for 15 min, washing cultures once and imaging them with an AMG EVOS FL microscope (*Video 1*).

## Microarray genotyping and PluriTest

RNA from passage ≥15 iPSCs was extracted using the Qiagen RNeasy kit according to the manufacturer's instructions. Quality of the extracted RNA was assessed using an Agilent (Santa Clara, California) Bioanalyzer 2100 (RIN scores for all samples ranged from 9.9 to 10), and RNA was processed into biotinylated cRNA and hybridized to the HT12v4 array using standard Illumina (San Diego, California) reagents as directed by the manufacturer. Arrays were scanned using an Illumina HiScan, and data processed using Illumina's GenomeStudio software. Using these data, we carried out PluriTest as previously described (*Müller et al., 2011*). Additionally, we mapped all detected HT12v4 probe sequences (n = 46,297) to the chimpanzee (panTro3) genome using BWA 0.6.3 (*Li and Durbin, 2009*). Probes that mapped to a single genomic location with no mismatches were retained (n = 21, 320, 46.2% of all probes) for the analysis that was restricted only to the chimpanzee lines.

When we considered data from human and chimpanzee iPSCs together, without excluding probes based on sequence matches to the chimpanzee genome, all chimpanzee lines in the panel had pluripotency scores slightly below the pluripotency threshold (*Figure 3—figure supplement 1*, lighter points). However, low pluripotency scores could stem from differences in our ability to estimate gene expression levels in the chimpanzee compared to the human due to attenuated hybridization caused by sequence divergence (*Gilad et al., 2005*). Indeed, when we subset the array to retain only those detected probes that map to the chimpanzee genome with no ambiguity or mismatches, all chimpanzee lines have pluripotency scores greater than the pluripotency threshold value of 20 (*Figure 3—figure supplement 1*, darker points).

## RNA sequencing and differential expression testing between iPSCs

50 bp single-end RNA sequencing libraries were generated from RNA extracted from 7 chimpanzee and 7 human iPSC lines using the Illumina TruSeq kit as directed by the manufacturer, as well as from their precursor fibroblast or LCL cell lines. All iPSC samples were multiplexed and sequenced on four lanes of an Illumina HiSeq 2500; while the precursor cell lines were multiplexed and sequenced on six lanes of the same sequencer. We generated a minimum of 28,010,126 raw reads per sample (*Supplementary file 6*), and confirmed the raw data were of high quality using FastQC (available online at http://www.bioinformatics.babraham.ac.uk/projects/fastqc/). We mapped raw reads to the chimpanzee (panTro3) or human (hg19) genome as appropriate using TopHat 2.0.8 (*Trapnell et al., 2009*), allowing for a maximum of 2 mismatches in each read. Due to the relatively poor annotation of the chimpanzee genome and to prevent biases in expression level estimates due to differences in mRNA transcript size and genetic divergence between the two species, we limited the analysis to reads that mapped to a list of orthologous metaexons across 30,030 Ensembl genes drawn from hg19 and panTro3, as in *Blekhman et al. (2010)*. Following mapping, gene level read counts were generated using *featureCounts* 1.4.4 as implemented in Subread (*Liao et al., 2013*). Due to mapping biases between human and chimpanzee ribosomal proteins and pseudogenes, we removed all genes associated with the Gene Ontology Cellular Compartment category 'ribosome' (GO:0005840, n = 141) and all annotated pseudogenes in Ensembl release 65 (n = 3170, December 2011, the oldest available archival version of Ensembl) from the data at this point.

We considered two normalization approaches in our analysis. In one instance, we examined only RNA-sequencing data from chimpanzee and human iPSCs, and retained 12,171 genes with at least 4 observations in one of the two species of $\log_2$ CPM > 1. CPM were then loess normalized by species within individuals with *voom* (*Law et al., 2014*). As the orthologous genes are not constrained to be

the same length in both species, we computed RPKM for each gene before carrying out any inter-species comparisons. We then used the R/Bioconductor package *limma* 3.20.3 (*Smyth, 2004*) to test for differential expression in our RNA-seq data, with a model that included only a species effect. Finally, we tested for an enrichment of GO categories amongst DE genes using the R package *topGO* 2.16.0 (*Alexa et al., 2006*). These normalised values were used only to identify genes DE between iPSCs of the two species.

For the dataset containing RNA-sequencing data from iPSCs and their precursors, we again only retained 13,147 genes with at least 4 observations in one of the four groups (chimpanzee iPSCs, chimpanzee precursors, human iPSCs or human precursors) of $\log_2$ CPM > 1. Gene counts were then loess normalised within individuals by tissue, after correcting for the lack of independence within different tissues from the same individual, through the function corfit. As above, we then computed species-specific RPKM values, and used *limma* and *topGO* to test for differential expression and GO category enrichment, respectively. In this instance, we used a model design with 6 parameters for the main effect (chimpanzee iPSC, human LCL-derived iPSC, human fibroblast-derived iPSC, chimpanzee fibroblast, human LCL and human fibroblast) and no additional covariates.

To confirm that our conclusions are robust with respect to the choice of normalization procedure, in both cases, we also tried a variety of other normalization schemes, including correcting for %GC content as in *Risso et al. (2011)*, none of which had a substantial effect on the final results (*Supplementary file 6*). Finally, we built neighbor joining trees using Manhattan distances calculated from RPKM values at all 13,147genes using the nj function in the R library *ape* (*Paradis et al., 2004*). All analyses were performed at a false discovery rate (*Benjamini and Hochberg, 1995*) threshold of 1% unless otherwise noted, using R 3.1.0 (*R Development Core Team, 2013*) and Bioconductor 2.14 (*Gentleman et al., 2004*).

## DNA methylation arrays

To analyze DNA methylation, we extracted DNA from all chimpanzee and human iPSC lines described above, as well as from the source fibroblast or LCLs. In all cases, 1000 ng of genomic DNA were bisulphite-converted and hybridized to the Infinium HumanMethylation450 BeadChip at the University of Chicago Functional Genomics facility as directed by the manufacturer. Since the probes on the array were designed using the human reference genome, we followed the approach described in *Hernando-Herraez et al. (2013)* to compare humans and chimpanzees. We retained those probes that had either a perfect match to the chimpanzee reference genome, or had 1 or 2 mismatches in the first 45 bp but no mismatches in the 3′ 5 bp closest to the CpG site being assayed. We also removed all probes that contained human SNPs (MAF $\geq$0.05) or chimpanzee SNPs (MAF $\geq$0.15) within the last 5 bp of their binding site closest to the CpG being assayed. Within each individual, probes with a detection p > 0.01 were excluded. This resulted in the retention of 335,307 autosomal probes, and an additional 8210 X chromosome probes, which we normalized and analyzed separately by sex. In all cases we performed a two-color channel signal adjustment, quantile normalization and β-value recalculation as implemented in the lumi package (*Du et al., 2008*). Because the HumanMethylation450 BeadChip contains two assay types which utilize different probe designs, we performed a BMIQ (beta mixture quantile method) normalization (*Teschendorff et al., 2013*) on the quantile-normalized autosomal data set. We did not perform this step on the X chromosome data, due to its methylation patterns. We built neighbor joining trees using Manhattan distances at all 335,307 probes using the nj function as above.

In order to identify DM probes we used an identical approach to that described above for the identification of DE genes. First, we identified probes that were DM between the iPSCs of both species using *limma* by using a reduced data set and model containing only data from the iPSCs themselves. Then, we fit a linear model to the data using limma with 6 parameters corresponding to the 6 tissue/species combinations in the data, classifying probes as DM at an FDR of 1%. As with the expression data, the reduced model has more power to identify DM probes between the two iPSC groups than the full model; however, there is great concordance between the two sets of results (*Figure 7—figure supplement 5*). We excluded all probes with mean β inter-group differences <0.1 in order to group DM probes into DMRs, which we define as 2 or more DM probes separated by <1 kb, with the additional requirement that the effect be in the same direction in all DM probes within the region. Finally, to examine the content of these DMRs, we used annotation files for the HumanMethylation450

Bead Chip provided by the manufacturer and discarded all DMRs associated with either multiple or no genes. We tested for enrichment of GO BP categories amongst the genes contained in the DMRs by using the R package topGO 2.16.0 (*Alexa et al., 2006*), using as a background set all genes in which it is theoretically possible to detect DMRs.

## H3K27ac and H3K27me3 ChIP-seq data

ChIP-seq assays were performed as previously described (*Schmidt et al., 2009*), with slight modifications. Specifically, approximately 60 million iPSCs from three chimpanzee individuals (C2, C5 and C7) were cross-linked with 1% formaldehyde for 10 min. Cells were lysed and chromatin sheared with a Covaris S2 (settings: 4 min, duty cycle 10%, 5 intensity, 200 cycles per burst in 4 6 × 16 mm tubes per individual). H3K27ac- and H3K27me3-enriched regions were isolated using 5 μg of either H3K27ac antibody (ab4729, Abcam, Cambridge, MA, USA) or H3K27me3 antibody (07–449, Millipore, Billerica, MA, USA). ChIP and input DNA from each individual were end-repaired, A-tailed and ligated to Illumina Truseq sequencing adapters before 18 cycles of PCR amplification. 200–300 bp DNA fragments were selected for sequencing. Input libraries were multiplexed and sequenced on one lane of an Illumina HiSeq2500 using the rapid run mode, ChIP libraries were multiplexed and sequenced on three lanes of an Illumina HiSeq2500 using the rapid run mode.

For comparison purposes, we downloaded ChIP input, H3K27ac and H3K27me3 data from 3 human iPSC lines (iPS 6.9, iPS-18a, and iPS.20b, all of them release 5) generated by the Roadmap Epigenomics Consortium (*Roadmap Epigenomics Consortium et al., 2015*) from the NIH GEO database (*Supplementary file 6*). Human and chimpanzee samples were mapped to either hg19 or panTro3 using BWA 0.7.9 (*Li and Durbin, 2009*); reads that mapped outside chromosomes 1–22 + X were discarded, as were reads that did not map uniquely to a single genomic region with less than 2 mismatches, or reads that were marked by Picard (http://picard.sourceforge.net) as originating from PCR duplicates.

After mapping and filtering, we used MACS 1.4.4 (*Zhang et al., 2008*) and RSEG 0.4.4 (*Song and Smith, 2011*) to identify peaks in the H3K27ac and H3K27me3 data respectively. Our analyses in this section follow those of *Zhou et al. (2014)*. Briefly, for MACS, we specified an initial p-value threshold of H3K27ac, 0.001, and used each line's ChIP input file for comparison. For RSEG, we used the 'rseg-diff' function to compare H3K27me3 enrichment against each individual's ChIP input file, with the recommended 20 maximum iterations for hidden Markov model training. We then filtered enriched regions or peaks identified by either program by retaining only those that overlapped a previously defined set of 200 bp orthologous windows (*Zhou et al., 2014*), where at least 80% of bases are mappable across species using liftOver. We define mappability as the ability of each 20 bp kmer beginning in that window to be uniquely mapped to the genome.

To ensure that sequence divergence did not confound our analyses, we mapped each identified region or peak in humans to the chimpanzee genome, and vice versa, using liftOver, and excluded regions and peaks where 80% or greater of bases in the enriched peaks or regions failed to align to the other genome. To further minimise the number of false positive results in our interspecies comparison (due to incomplete power), we applied a two-step cutoff (*Cain et al., 2011*) to the list of enriched regions and peaks. For H3K27ac, we retained all peaks that were identified with a first, stringent cutoff of FDR < 5% in one species and a, second, relaxed cutoff of FDR < 15% in the other, as in *Zhou et al. (2014)*. Because RSEG does not report FDR values for enriched regions, we used each region's domain score, which is the sum of the posterior scores of all bins within the domain, and set a first, stringent cutoff of 20 in one species, and a second, relaxed threshold demanding only that the region be classified as 'enriched' by RSEG, without a specific score requirement.

Having done this, we integrated data from multiple peaks (when present) to generate a gene-level metric of ChIP signal in each individual. Specifically, we computed an enrichment score for each histone mark in each individual in a set of previously defined 26,115 orthologous TSSs (*Zhou et al., 2014*) by dividing RPKM values at each TSS at gene *i* for either mark minus RPKM values in TSS at gene *i* for ChIP input, all of it over the genome-wide average RPKM for either mark minus the genome-wide average RPKM for ChIP input. Given the way in which we have defined this enrichment score, a score >0 indicates those genes where we detected more histone mark reads than input reads, while a score >1 indicates a gene with an excess of histone mark reads greater than what we would expect given the genome-wide distribution.

Because 8 of the 22 genes in the list of pluripotency master genes used to generate *Figure 5D* do not have clearly defined orthologous TSSs, we also examined whether MACS identified peaks in

the 2 kb ± TSS for all 22 genes and their orthologous position in the chimpanzee genome, identified solely through liftOver—that is, without taking into account whether there is evidence for a TSS at that position in the chimpanzee genome. To generate *Figure 5—figure supplement 3*, we simply asked how many of the 22 genes had at least 1 peak at an FDR < 5% in at least 1 individual in either species, regardless of orthology and sequence conservation.

We note that since different labs produced the human and chimpanzee data, we expect a considerable technical batch effect to be completely confounded with species annotation. Given this study design, we expect the technical batch effect to result in the appearance of inter-species differences; yet, our goal is to demonstrate similarity across species. Thus, our conclusions (of high overlap across species), are conservative with respect to the technical batch effect.

## REX1 expression and function

To examine the possible consequences of reduced expression of *REX1* in chimpanzee iPSCs, we retrieved genes that responded to a *REX1* knockdown in mESCs from *Supplementary files 2, 3* of *Scotland et al. (2009)* and converted Affymetrix MG 430 2.0 probe IDs to ENSM and ultimately to orthologous ENSG identifiers using Biomart release 66 (to control for deprecated identifiers). Because *Masui et al. (2008)*; *Scotland et al. (2009)*; *Son et al. (2013)* have highlighted *REX1*'s function in controlling cell cycle progression, glycolysis and cellular differentiation, we additionally retrieved genes associated with these terms to generate *Figure 6C* as follows: The core set of pluripotency TFs are those described by *Orkin and Hochedlinger (2011)* and *Ng and Surani (2011)*. Cell cycle and glycolysis categories contain all genes associated with GO BP:0007049 and BP:0006096 respectively, whereas cell fate contains genes associated with any GO term that contains the words 'ectoderm', 'mesoderm' or 'endoderm'. We also examined individual examples of cell fate differentiation: CNS development genes are associated with BP:0007417 or any of its offspring; cardiovascular system development genes are associated with BP:0072358 or any of its offspring; hepatobiliary system development genes are associated with BP:0055123 or any of its offspring. Confidence intervals around the null hypothesis were generated independently for each category from 100,000 permutations in R.

Finally, to compare our data with a previously published set of chimpanzee and bonobo iPSC lines (*Marchetto et al., 2013b*), we downloaded fastq files from GEO (Series GSE47626) and mapped only the first mate from all reads using the same approach as above, but allowing 4 mismatches in the entire 100 bp read. We normalised expression estimates jointly with our own cell lines to generate *Figure 6—figure supplement 2*, and used all data points from a given species, irrespective of origin, to generate boxplots.

To generate *figure 6—figure supplement 3*, we took Illumina HT12v4 array data from 73 human iPSC lines generated in house and calculated Pluritest pluripotency and novelty scores as above, using the full probe set. Independently, we normalised and background-corrected the raw array intensities using the lumiExpresso function in *lumi* and extracted expression values for all 73 human iPSC lines at the single array probe associated with *REX1*.

In order to examine *REX1* methylation levels in other human PSCs, we obtained Infinium HumanMethylation450 BeadChip for the lines reported in *Ziller et al. (2011)* from the authors, and normalised it jointly with our own data as described above. We then extracted normalised methylation β levels at the 13 probes that map to *REX1* in both chimpanzees and humans to generate *Figure 6—figure supplement 4*.

## Other indicators of genomic stability

Finally, we assessed two broad indicators of stability in our chimpanzee lines. All iPSC lines derived from female chimpanzees, and 3 of 4 lines derived from human females, show strong evidence for elevated expression of XIST relative to male lines (FDR-adjusted p = 0.0010; *Figure 7—figure supplement 6*) and maintenance of X-chromosome inactivation during pluripotency. X-chromosome methylation patterns in females corroborate these observations, with the majority of probes mapping to the X-chromosome in our data being either hemimethylated ($0.2 < \beta < 0.8$) or hypermethylated ($\beta \geq 0.8$) in females but not in males (*Figure 7—figure supplement 7*). We also used a list of 168 imprinted probes from *Ma et al. (2014)* to check for maintenance of genomic imprinting after reprogramming. We find that the majority of imprinted loci remain hemimethylated following reprogramming in both human and

chimpanzee iPSC lines (*Figure 7—figure supplement 8*). However, we identify two sets of probes that are consistently hypermethylated in pluripotent lines but were hemimethylated in their precursor cells. The first cluster contains 5 probes that are hypermethylated across both chimpanzee and human iPSCs; these probes are associated with the genes *KCNK9*, *ANKRD11* and *MKRN3*. The second cluster is comprised of 21 probes that are hypermethylated in all human iPSCs but only 2 chimpanzee iPSCs in our data, and is associated with the gene *PEG3-ZIM2*, which has been previously shown to be abnormally methylated in both hESCs and hiPSCs (*Lund et al., 2012*).

## Data Access

All novel RNA-sequencing, DNA methylation and ChIP-seq data are available at the GEO under SuperSeries number GSE61343. Additionally, a table with p-values for all hypothesis testing performed using the methylation data (by probe) is available on the Gilad lab website (http://giladlab.uchicago.edu/Data.html).

All chimpanzee iPSC lines described in this publication are available fully and without restrictions to other investigators upon request to the corresponding authors.

## Acknowledgements

We thank members of the Gilad and Marques-Bonet labs for helpful discussions; Julien Roux for help with differential gene expression analysis; Alexander Meissner for sharing methylation data from human iPSC and ESC lines, Fred Gage for providing a list of differentially expressed genes in retrovirally-reprogrammed chimpanzee iPSCs, and an anonymous reviewer for suggesting that we add comparative ChIP-seq data from H3K27ac and H3K27me3. This work was supported by NIH grant GM077959 to YG as well as by grants from the California Institute for Regenerative Medicine (CIRM) CL1-00502 and TR01250 to JFL, and ERC Starting Grant 260372 and MICINN (Spain) BFU2011-28549 to TM-B. IGR is supported by a Sir Henry Wellcome Postdoctoral Fellowship; IHH is supported by FI Generalitat Catalunya; MCW is supported by an EMBO Long-Term Fellowship (ALTF 751-2014) and the European Commission Marie Curie Actions; NEB is supported by an NIH training grant (GM007197) and an NIH pre-doctoral award (F31 AG 044948); LCL is supported by a WRHR Career Development award (NIH K12 HD001259) and the UCSD Department of Reproductive Medicine; TRL and KS are supported by the UCSD Department of Reproductive Medicine; CLK is supported by a CTSA TL1 pre-doctoral fellowship (TR 432-7). We also acknowledge the generous support of the Yerkes Primate Center through their grant ORIP/OD P51OD011132.

## Additional information

### Funding

| Funder | Grant reference | Author |
|---|---|---|
| Wellcome Trust | Sir Henry Wellcome Postdoctoral Fellowship | Irene Gallego Romero |
| National Institutes of Health (NIH) | GM077959 | Yoav Gilad |
| California Institute for Regenerative Medicine (CIRM) | CL1-00502 and TR01250 | Jeanne F Loring |
| European Research Council (ERC) | Starting Grant 260372 | Tomas Marques-Bonet |
| Ministerio de Economía y Competitividad | BFU2011-28549 | Tomas Marques-Bonet |
| University of California, San Diego, Department of Reproductive Medicine | | Louise C Laurent, Trevor R Leonardo, Karen Sabatini |
| Generalitat de Catalunya | | Irene Hernando-Herraez |
| National Institutes of Health (NIH) | Training Grant GM007197 and F31 AG044948 | Nicholas E Banovich |

| Funder | Grant reference | Author |
|---|---|---|
| National Institutes of Health (NIH) | K12 HD001259 | Louise C Laurent |
| EMBO | ALTF 751-2014 | Michelle C Ward |
| National Center for Advancing Translational Sciences | Clinical and Translational Science Awards, TL1 pre-doctoral fellowship TR 432-7 | Courtney L Kagan |

The funders had no role in study design, data collection and interpretation, or the decision to submit the work for publication.

## Author contributions

IGR, BJP, YG, Conception and design, Acquisition of data, Analysis and interpretation of data, Drafting or revising the article, Contributed unpublished essential data or reagents; IH-H, XZ, Analysis and interpretation of data, Drafting or revising the article; MCW, NEB, CLK, JEB, CHH, AM, CIC, Acquisition of data, Drafting or revising the article; IFB-N, Conception and design, Acquisition of data, Drafting or revising the article, Contributed unpublished essential data or reagents; YL, KS, TRL, MP, Acquisition of data, Analysis and interpretation of data, Drafting or revising the article; TM-B, Conception and design, Analysis and interpretation of data, Drafting or revising the article; LCL, JFL, Conception and design, Acquisition of data, Analysis and interpretation of data, Drafting or revising the article

## Author ORCIDs

Irene Gallego Romero, http://orcid.org/0000-0003-1613-8998

# Additional files

## Supplementary files

• Supplementary file 1. Description of samples. a: Descriptive data for all chimpanzee cell lines used in this work. b: Descriptive data for all human iPSC lines used.

• Supplementary file 2. Origin and purpose of all primers used.

• Supplementary file 3. Normalized RPKM values and DE genes between chimpanzee and human iPSCs.

• Supplementary file 4. Gene Ontology BP terms associated with genes DE between chimpanzee and human iPSCs.

• Supplementary file 5. DMRs identified between chimpanzee and human iPSCs.

• Supplementary file 6. Genome-wide data summary statistics. a: Numbers of DM probes and DMRs between chimpanzee and human iPSCs identified under various mean β difference thresholds. b: RNA- and ChIP-sequencing reads generated and mapped for all samples in this work. c: Effects of different normalization schemes on the number of genes classified as DE in the full data set.

• Supplementary file 7. Gene Ontology BP terms associated with genes within DMRs between chimpanzee and human iPSCS.

• Supplementary file 8. H3K27ac and H3K27me3 enrichment scores in 3 chimpanzee and 3 human iPSCs around 26,115 orthologous TSSs.

• Supplementary file 9. Correlations between principal components and selected covariates. a: in the expression data. b: in the methylation data.

• Supplementary file 10. Normalized RPKM values and DE genes identified under the full *limma* DE testing framework.

• Supplementary file 11. DMRs identified between chimpanzee iPSCs and their precursor fibroblasts.

• Supplementary file 12. DMRs identified between human iPSCs and their precursor cells.

### Major datasets

The following datasets were generated:

| Author(s) | Year | Dataset title | Dataset ID and/or URL | Database, license, and accessibility information |
|---|---|---|---|---|
| Gallego Romero I, Pavlovic BJ, Hernando-Herraez I, Banovich NE, Kagan CL, Burnett JE, Huang CH, Mitrano A, Chavarria CI, Ben-Nun IF, Li Y, Sabatini K, Leonardo TR, Parast M, Marques-Bonet T, Laurent LC, Loring JF, Gilad Y | 2014 | Generation of a Panel of Induced Pluripotent Stem Cells From Chimpanzees: a Resource for Comparative Functional Genomics (RNA-Seq) | http://www.ncbi.nlm.nih.gov/geo/query/acc.cgi?acc=GSE60996 | Publicly available at the NCBI Gene Expression Omnibus (Accession no: GSE60996). |
| Gallego Romero I, Pavlovic BJ, Hernando-Herraez I, Banovich NE, Kagan CL, Burnett JE, Huang CH, Mitrano A, Chavarria CI, Ben-Nun IF, Li Y, Sabatini K, Leonardo TR, Parast M, Marques-Bonet T, Laurent LC, Loring JF, Gilad Y | 2014 | Generation of a Panel of Induced Pluripotent Stem Cells From Chimpanzees: a Resource for Comparative Functional Genomics (methylation array) | http://www.ncbi.nlm.nih.gov/geo/query/acc.cgi?acc=GSE61342 | Publicly available at the NCBI Gene Expression Omnibus (Accession no: GSE61342). |
| Gallego Romero I, Pavlovic BJ, Hernando-Herraez I, Zhou X, Ward MC, Banovich NE, Kagan CL, Burnett JE, Huang CH, Mitrano A, Chavarria CI, Ben-Nun IF, Li Y, Sabatini K, Leonardo TR, Parast M, Marques-Bonet T, Laurent LC, Loring JF, Gilad Y | 2015 | Generation of a Panel of Induced Pluripotent Stem Cells From Chimpanzees: a Resource for Comparative Functional Genomics | http://www.ncbi.nlm.nih.gov/geo/query/acc.cgi?acc=GSE69919 | Publicly available at the NCBI Gene Expression Omnibus (Accession no: GSE69919). |

The following previously published datasets were used:

| Author(s) | Year | Dataset title | Dataset ID and/or URL | Database, license, and accessibility information |
|---|---|---|---|---|
| Marchetto MC, Narvaiza I, Denli AM, Benner C, Gage FH | 2013 | Differential LINE-1 retrotransposition in induced pluripotent stem cells between humans and great apes | http://www.ncbi.nlm.nih.gov/geo/query/acc.cgi?acc=GSE47626 | Publicly available at the NCBI Gene Expression Omnibus (Accession no: GSE47626). |
| Roadmap Epigenomics Consortium Kundaje, Meuleman A, Ernst W, Bilenky J, Yen M, Heravi-Moussavi A, Kheradpour A, Zhang P, Wang JZ *et al* | 2015 | NIH Epigenomics Roadmap Initiative | http://www.ncbi.nlm.nih.gov/bioproject/PRJNA34535 | Publicly available at the NCBI Gene Expression Omnibus (Accession no: PRJNA34535). |

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
