## [Decision Letter]

Thank you for sending your work entitled “A Panel of Induced Pluripotent Stem Cells From Chimpanzees: a Resource for Comparative Functional Genomics” for consideration at *eLife*. Your Tools and resources article has been favorably evaluated by Fiona Watt (Senior editor), Duncan Odom (Reviewing editor), and two anonymous reviewers.

The Reviewing editor and the reviewers discussed their comments before we reached this decision, and the Reviewing editor has assembled the following comments to help you prepare a revised submission.

All reviewers felt that a panel of iPS cells from chimpanzee has real potential to be a valued Resource across the community, although there were substantial concerns that must be addressed before publication in *eLife*. Reviews abridged to elaborate specific points are found at the end of this note.

Required for publication:

1) In line with *eLife's* role as a premier open-access venue for high-impact science, all the reported iPS cell lines must be deposited for unrestricted access by any member of the community into ATCC (preferably) or an alternative, easily accessed depository.

2) Demonstration that these lines can be differentiated into neuronal cells (or an alternative differentiation pathway).

3) A few more molecular characterizations are required, including (a) adding a comparison with publicly available mouse iPS gene expression data and (b) addition of representative chromatin state experiments, in part to demonstrate activation of pluripotency genes.

4) Clarifications required. (a) *REX1* results were highlighted by two independent reviewers, below, as confusing. (b) Differentially expressed (and methylated) genes should be checked for functional pathway enrichments. (c) Degree of gene expression divergence is surprising and could be elaborated on.

Reviewer 1:

These cells need to be released to a public depository such as ATCC for widest community adoption. The most important influence publishing this manuscript would have, particularly in an open-access journal, is if many research groups adopt the Gilab lab's cells as the community standard for inter-primate functional genomics. This cannot occur unless the cells are accessible. Such easy access is a key selling point for publishing a resource in *eLife*. However, the discussion makes it appear that the authors reserve the right whether or not to release the cell lines: “Crucially for this, all lines from the panel are readily available for sharing with investigators who may wish to address questions beyond the scope of our own research” (quote from the Discussion).

Reviewer 2:

1) The major value of these chimpanzee iPSCs is that they can be differentiated into defined cell types that could not normally be obtained from live animals. The chimp iPSCs appear to be pluripotent based on the embryoid body assays and gene expression analyses described in the paper. However, can they be differentiated into any other cell types of interest (e.g., neuronal progenitors or neurons)? It would go a long way towards establishing the value of the resource if the authors could demonstrate this.

2) The degree of gene expression divergence between human and chimp iPSCs is surprising: about 38% of genes that could be measured show significant differential expression. Are many of these low-expressed genes that show minor fluctuations in their expression levels? Do genes that show high expression in one species and low expression in the other coalesce into particular biological categories (the authors report no GO enrichments for DE genes overall)? It is a bit worrisome that the chimp and human iPSCs show such extensive differences in expression, since it may suggest a species-specific response to the (highly artificial) iPSC protocol rather than intrinsic differences.

3) Including expression data from mouse iPSCs may help put the human-chimp expression differences into better context. We would expect gene expression patterns in the primate iPSCs to be more similar to each other than either would be to mouse. I understand there may be some limitations to this analysis, notably derivation artifacts in mouse iPSCs not generated in the authors' own lab using their protocols. However, if chimp iPSCs turn out to be outliers compared to human and mouse, that would warrant more investigation.

4) The connection between the differential methylation analysis and differential expression analysis is weak, in that DMR regions do not appear to correlate well with DE genes. However, the authors report ∼550 genes that show both DE and are associated with DMRs. What are the functions of these genes?

5) I would have liked to see some analyses of chromatin states in these cells, specifically histone modification profiles for H3K27me3 and H3K4me3 (or H3K27ac). We have clear expectations of what these profiles should look like in pluripotent cells: pluripotency genes should be highly active, and many fate commitment genes should be bivalent. It would also serve as an additional quality control, since iPSCs with aberrant chromatin states could be identified and excluded. Also, combining the DE and chromatin analyses would reveal important species differences—genes that are strongly differentially expressed between species, and that are associated with clear differences in activation state, would be especially interesting.

6) The *REX1* result is a bit confusing: this gene is required for pluripotency maintenance in mouse and human, but not chimp? Also, one of the chimpanzee lines clearly expresses the gene. Is the locus itself intact in all the chimpanzee lines? Alternatively, does high *REX1* expression in one cell line and low expression in the others suggest incomplete reversion to the pluripotent state in the low-expressing lines?

7) The authors compare gene expression in human and chimp fibroblast precursors and derived iPSCs. They identify 1,118 DE genes between the iPSCs—are there any functional enrichments here, especially in the subset of DE genes that are not differentially expressed between the human and chimp fibroblasts?

Reviewer 3:

The principal component analysis (PC) for both gene expression and methylation is just descriptive and does not add any new information (could go to a supplement).

The authors find that *REX1* is barely expressed in most of the generated chimpanzee iPSCells (6 out of 7). This finding is potentially interesting but incomplete. The main concern rises from the comparison with previously described studies as shown in Figure 6—figure supplement 2. The authors state: “Furthermore, and consistent with our observations, REX1 is either absent or expressed at low levels in both replicates of one of two retrovirally reprogrammed bonobo (*Pan paniscus*, sister species to chimpanzees) iPSC lines generated by (49) (Figure 6—figure supplement 2)”. This sentence is misleading since all the chimpanzee iPSCs (4 replicates) described by Marchetto el al. showed comparable levels of *REX1* expression with human iPSCs. What are the causes of this discrepancy, can it be due to the differences in reprograming? In addition, there are no follow up experiments showing the cause / consequences of lack of *REX* expression in chimpanzee iPSCs and further investigation of a potential compensation mechanism.

The authors should also clarify how they calculate the median for *REX* expression in chimpanzee iPSCs (Figure 6—figure supplement 2).

---

## [Author Response]

Reviewer 1:

*These cells need to be released to a public depository such as ATCC for widest community adoption. The most important influence publishing this manuscript would have, particularly in an open-access journal, is if many research groups adopt the Gilab lab's cells as the community standard for inter-primate functional genomics. This cannot occur unless the cells are accessible. Such easy access is a key selling point for publishing a resource in* eLife*. However, the discussion makes it appear that the authors reserve the right whether or not to release the cell lines: “Crucially for this, all lines from the panel are readily available for sharing with investigators who may wish to address questions beyond the scope of our own research” (quote from the Discussion)*.

We agree with the reviewer. We did not mean for this statement to be ambiguous. We now clearly state that these lines are available fully and without restrictions to other investigators upon request to the corresponding authors (Results and Discussion, subsection headed “Data Access”).

Reviewer 2:

*1) The major value of these chimpanzee iPSCs is that they can be differentiated into defined cell types that could not normally be obtained from live animals. The chimp iPSCs appear to be pluripotent based on the embryoid body assays and gene expression analyses described in the paper. However, can they be differentiated into any other cell types of interest (e.g., neuronal progenitors or neurons)? It would go a long way towards establishing the value of the resource if the authors could demonstrate this*.

It is perhaps worth noting that we are not aware of cases where iPSCs that were determined to be fully pluripotent failed to differentiate when directed methods were used (though, granted, different lines tend to differentiate to certain cell types more efficiently than others). That said, we now provide descriptions and the results of directed differentiations of chimpanzee iPSCs to hepatocytes and cardiomyocytes (please see the subsections “Characterizing the chimpanzee iPSCs” and “Directed differentiation of chimpanzee iPSCs to hepatocytes and cardiomyocytes”, and new Figure 2—figure supplement 2 and Figure 2—figure supplement 3).

*2) The degree of gene expression divergence between human and chimp iPSCs is surprising: about 38% of genes that could be measured show significant differential expression. Are many of these low-expressed genes that show minor fluctuations in their expression levels? Do genes that show high expression in one species and low expression in the other coalesce into particular biological categories (the authors report no GO enrichments for DE genes overall)? It is a bit worrisome that the chimp and human iPSCs show such extensive differences in expression, since it may suggest a species-specific response to the (highly artificial) iPSC protocol rather than intrinsic differences*.

This is an important concern, and we indeed did not provide sufficient discussion of this observation. The short answer is that due to overall lower variation in gene expression levels in iPSCs both within and between species, we actually have more power to detect even small inter-species regulatory differences.

Here is a longer response (we also revised the text accordingly; please see the subsection headed “Interspecies analysis of gene expression and methylation data from iPSCs”, the fifth paragraph of the Discussion, Figure 4—figure supplement 2, and [Supplementary-material SD5-data]).

We acknowledge that the fraction of genes we identify as DE between human and chimpanzee iPSCs is indeed greater than previously reported in other primary tissues, when studies considered similar sample sizes (e.g., [8], [7]). Yet, we believe that the reason for this difference is likely to be increased power to detect DE in our iPSC sample relative to previous work. Indeed, given the small amount of intra-species variability we observe in RNA-seq and methylation relative to other tissues, we expect to have greater power to detect small, statistically significantly inter-species differences that would have been missed by studies that consider more variable tissue samples. This hypothesis is supported by the observation that the fraction of genes with log FC < 2 we detect as DE between human and chimpanzee iPSCs is much *greater* than in other comparison we have performed with any other tissue (Figure 4—figure supplement 2).

Moreover, other than *REX1*, which is discussed at length in the text, we find no noteworthy differences between human and chimpanzee iPSCs in master pluripotency regulators genes (including in the new ChIP-seq data). These observations suggest that the core circuitry of pluripotency is behaving similarly in both species.

With regards to pathway enrichments, when we test for GO BP term enrichment amongst the 546 genes with an absolute log_2_ fold-change difference in interspecies mean expression levels > 2, we find 123 overrepresented terms ([Supplementary-material SD5-data]), none of which suggest (to us at least) a potential difference in pluripotency potential.

Therefore, we believe that that high number of DE genes we observed between human and chimpanzee iPSCs primarily reflects a combination of true biological differences and increased power to detect small differences in expression between the two species (that might not be of great biological significance).

Of note (though for obvious reasons we do not discuss this in the current manuscript): we have seen similar patterns when we map eQTLs in iPSCs compared with eQTLs in other tissues.

*3) Including expression data from mouse iPSCs may help put the human-chimp expression differences into better context. We would expect gene expression patterns in the primate iPSCs to be more similar to each other than either would be to mouse. I understand there may be some limitations to this analysis, notably derivation artifacts in mouse iPSCs not generated in the authors' own lab using their protocols. However, if chimp iPSCs turn out to be outliers compared to human and mouse, that would warrant more investigation*.

The probability that our chimp data would be an outlier when mouse data are considered is remote, even due to technical considerations alone.

Nevertheless, to address this point we examined data from Guo et al., Cell 156(4): 649–662, which contains RNA-seq data from 4 independently derived mouse iPSCs through lentiviral reprogramming of mouse embryonic fibroblasts. We began our analysis using the table of reads mapped to known mouse genes (mm9) by the authors using TopHat 2.0.4 (the data is available in GEO as series GSE53074) and used the R package biomaRt to retrieve mouse and human homologous identifiers for all genes in the Guo et al. data. We intersected those genes with those in our own data set, and calculated TMM library sizes, CPM and loess normalised data across 10,580 retained genes from all three species jointly. We then computed RPKM in a species-specific manner. Because the Guo et al. data are provided at the gene level, we used the mean length of all known transcripts associated with a particular Ensembl ID as the gene length to calculate RPKM for the mouse data.

As expected, both PCA and a neighbour-joining tree built from Manhattan distances (see Figure 8) confirm that data from the chimpanzee iPSC (purple, C1-C7 on the figure in the right) lines are far more similar to data from human iPSCs (yellow-green, H1-H7 on the figure in the right) than from mouse iPSCs (blue, M1-M4 on the figure in the right).

Author response image 1.**DOI:**
http://dx.doi.org/10.7554/eLife.07103.052

We respectfully ask not to include these results in the paper. Because strong batch effects are impossible to account for in this analysis, we do not believe that it is providing meaningful insight.

4) The connection between the differential methylation analysis and differential expression analysis is weak, in that DMR regions do not appear to correlate well with DE genes. However, the authors report ∼550 genes that show both DE and are associated with DMRs. What are the functions of these genes?

In response to the suggestion, we performed enrichment analyses for genes that were both DE and were associated with DMRs at a 0.1 and 0.3 mean β difference threshold, and in neither case found any significant enrichments for GO BP or MF terms after correcting for multiple testing. This has now been clarified in the text (in the subsection “Interspecies analysis of gene expression and methylation data from iPSCs”).

*5) I would have liked to see some analyses of chromatin states in these cells, specifically histone modification profiles for H3K27me3 and H3K4me3 (or H3K27ac). We have clear expectations of what these profiles should look like in pluripotent cells: pluripotency genes should be highly active, and many fate commitment genes should be bivalent. It would also serve as an additional quality control, since iPSCs with aberrant chromatin states could be identified and excluded. Also, combining the DE and chromatin analyses would reveal important species differences—genes that are strongly differentially expressed between species, and that are associated with clear differences in activation state, would be especially interesting*.

Following a discussion with the editor, we were asked to provide ChIP-seq for H3K27ac and H3K27me3 from two representative chimpanzee iPSC lines. Accordingly, we have collected and analysed new ChIP-seq data for these marks in three chimpanzee individuals and compared the patterns we observed in the chimpanzee iPSC lines to similar ChIP-seq data from humans (collected by the Roadmap Epigenomics project).

As we report in the subsection headed “Comparative histone modification data”, genome-wide overlap in the enrichment of these histone modifications across the two species is considerably greater than expected by chance. When we consider histone modifications in proximity to genes known to either be poised in human iPSCs, or important for pluripotency, the overlap in mark enrichments across the species is even higher.

These results are presented in the new main text, Figure 5, Figure 5—figure supplement 1, Figure 5—figure supplement 2 and Figure 5—figure supplement 3 and new [Supplementary-material SD9-data]. We have also added two authors to the manuscript, Xiang Zhou and Michelle C. Ward, to reflect their contributions towards generating and analysing these novel data. Raw and processed data is being deposited in GEO under SuperSeries number GSE61343.

*6) The* REX1 *result is a bit confusing: this gene is required for pluripotency maintenance in mouse and human, but not chimp? Also, one of the chimpanzee lines clearly expresses the gene. Is the locus itself intact in all the chimpanzee lines? Alternatively, does high* REX1 *expression in one cell line and low expression in the others suggest incomplete reversion to the pluripotent state in the low-expressing lines?*

The short answer to the last question is: no, we have not seen any evidence to suggest incomplete reprograming. More generally, we agree with this comment – this result was not reported very clearly.

In response, we revised the text and added significantly to this section (“*REX1* may be dispensable for chimpanzee pluripotency“, revised Figure 6, and new Figure 6—figure supplement 3 and Figure 6—figure supplement 4). Briefly, we added the following data and results:

No large-scale expression differences between *REX1* expressing chimpanzee iPSC line C6 and all other chimpanzee lines, even in genes hypothesised to be directly regulated by *REX1* in humans (see the new panels C and D of Figure 6). Evidence of variable expression and methylation in *REX1* in human iPSC lines, but not in ESCs, although at a much lower frequency than in chimpanzees (see new Figure 6—figure supplement 3 and Figure 6—figure supplement 4.

*7) The authors compare gene expression in human and chimp fibroblast precursors and derived iPSCs. They identify 1,118 DE genes between the iPSCs—are there any functional enrichments here*, *especially in the subset of DE genes that are not differentially expressed between the human and chimp fibroblasts?*

We found no significant functional enrichments. We have now clarified this (“None of these gene sets were significantly enriched for functional annotations using GO BP terms”).

Reviewer 3:

*The principal component analysis (PC) for both gene expression and methylation is just descriptive and does not add any new information (could go to a supplement)*.

We agree that these figures are primarily descriptive, but given the sections in the manuscript in which the data they contain is presented, we believe they are best presented as independent main text figures rather than figure supplements or as additional panels in Figure 1.

*The authors find that* REX1 *is barely expressed in most of the generated chimpanzee iPSCells (6 out of 7). This finding is potentially interesting but incomplete. The main concern rises from the comparison with previously described studies as shown in*
Figure 6—figure supplement 2*. The authors state: “Furthermore, and consistent with our observations,* REX1 *is either absent or expressed at low levels in both replicates of one of two retrovirally reprogrammed bonobo (*Pan paniscus, *sister species to chimpanzees) iPSC lines generated by (*[49]*) (*Figure 6—figure supplement 2*)”. This sentence is misleading since all the chimpanzee iPSCs (4 replicates) described by Marchetto el al. showed comparable levels of* REX1 *expression with human iPSCs. What are the causes of this discrepancy, can it be due to the differences in reprograming? In addition, there are no follow up experiments showing the cause / consequences of lack of* REX *expression in chimpanzee iPSCs and further investigation of a potential compensation mechanism.*

Thank you, these are good points. We have addressed many of the reviewer's broader concerns above, in our response to reviewer 2's 6th point. With regards to specific question/concerns about the text, we have clarified the sentence highlighted as problematic by the reviewer as follows:

“Furthermore, and consistent with our observations, *REX1* is either absent or expressed at low levels in both replicates of one of two retrovirally reprogrammed bonobo (*Pan paniscus*, sister species to chimpanzees) iPSC lines generated by (49) although it is expressed in both replicates of both chimpanzee iPCS lines from the same group (Figure 6—figure supplement 2).”

We are currently pursuing a more in-depth investigation of the causes and consequences of this difference in *REX1* expression levels, but we believe presenting in-depth experimental validation additional to the analyses already included is beyond the scope of this paper, and hope the reviewer agrees.

*The authors should also clarify how they calculate the median for* REX *expression in chimpanzee iPSCs (*Figure 6—figure supplement 2*)*.

The reported median expression values in this figure were calculated using all data points from a given species, as indicated in the figure legend and described in the Methods.